# SERTM2: a neuroactive player in the world of micropeptides

Michela Lisi [1,2], Tiziana Santini[1,2], Tiziano D'Andrea[3], Beatrice Salvatori[2], Adriano Setti[1], Alessandro Paiardini[4], Sofia Nutarelli [1], Carmine Nicoletti [5], Flaminia Pellegrini [1], Sergio Fucile [3,6], Irene Bozzoni [1,2,7] & Julie Martone [8]

## Abstract

In this study, we analyze the long noncoding RNA, lncMN3, that is predominantly expressed in motor neurons and shows potential coding capabilities. Utilizing custom antibodies, we demonstrate the production of a lncMN3-derived type I transmembrane micropeptide, SERTM2. Patch-clamp experiments performed on both wild-type and SERTM2 knockout motor neurons, differentiated in vitro from mouse embryonic stem cells, show a difference in the resting membrane potential and overall decreased excitability upon SERTM2 depletion. In vivo studies indicate that the absence of the peptide impairs treadmill test performance. At the mechanistic level, we identify a two-pore domain potassium channel, TASK1, known to be a major determinant of the resting membrane potential in motor neurons, as a SERTM2 interactor. Our study characterizes one of the first lncRNA-derived micropeptides involved in neuronal physiology.

**Keywords** Long Noncoding RNAs; Micropeptides; Motor Neurons; Potassium Ion Channels; TASK1
**Subject Categories** Membranes & Trafficking; Neuroscience; RNA Biology

See also: F-Y Hsu et al

## Introduction

Since the discovery of their functional role, long noncoding RNAs have been defined as noncoding RNAs with a length greater than 200 nucleotides, a value established arbitrarily and based on experimental reasons related to both biochemical and biophysical methodologies. Other distinguishing features of lncRNAs reside, with a few exceptions (e.g., MALAT1 and NEAT1), in a greater specificity of expression, both temporal and spatial, a greater evolutionary speed and a lower level of expression when compared to coding RNAs (Ulitsky, 2016; Statello et al, 2021). This view has recently been revised, and it has been proposed to increase the size limit to more than 500 nt (Mattick et al, 2023); in addition, even the generic definition of noncoding transcripts has taken on new facets as a dual role has been identified for several lncRNAs: one fulfilled by the RNA molecule itself and the other, characteristic of the ones located in the cytoplasm, by the ability to produce micropeptides (van Heesch et al, 2019; Ji et al, 2015; Pan et al, 2022). The translational hypothesis is also supported by their structure, which is very similar to that of mRNAs, as they are transcribed mainly by RNA polymerase II, generally have a 7-methylguanosine triphosphate cap at the 5' end and are polyadenylated at the 3' end (Chew et al, 2013; Statello et al, 2021). Moreover, a large fraction of lncRNAs have been found associated with ribosomes, and it has been speculated that there may be many undiscovered small peptides encoded within lncRNAs (Nelson et al, 2014). An overwhelming confirmation of this speculation was obtained years later by Chen et al (2020) that identified a large subset of lncRNAs able to encode for microproteins with critical roles in different cellular pathways and by Duffy et al (2022) that described the production of micropeptides from lncRNAs induced during human neuronal activity. A role in de novo protein evolution has also been suggested (Ruiz-Orera et al, 2014).

Some lncRNA-derived micropeptide have been found deregulated in cancers, such as the case of the 21-amino-acid survival-associated micropeptide XBP1SBM that is encoded by the lncRNA MLLT4-AS1 and resulted to be upregulated in triple-negative breast cancer (TNBC) (Wu et al, 2022) or of the 53-aa-long conserved peptide HOXB-AS3 involved in the suppression of colon cancer cell growth, colony formation, migration, invasion and tumorigenesis (Huang et al, 2017).

From a physiological point of view, one of the first tissues in which their role has been studied in more detail is the muscle (Bi et al, 2017; Makarewich et al, 2018; Lin et al, 2019; Shi et al, 2017; Quinn et al, 2017; Zhang et al, 2017). Pioneering studies from the Olson's group on transcripts that were previously annotated as lncRNAs identified two muscle-specific transmembrane micropeptides, Myoregulin (MLN, Anderson et al, 2015) and Dwarf open reading frame (DWORF; Nelson et al, 2016). These micropeptides

[1]Department of Biology and Biotechnologies "Charles Darwin", Sapienza University of Rome, Rome, Italy. [2]Center for Life Nano-& Neuro-Science, Fondazione Istituto Italiano di Tecnologia, Rome, Italy. [3]IRCCS Neuromed, Pozzilli, Italy. [4]Department of Biochemical Sciences, Sapienza University of Rome, Rome, Italy. [5]DAHFMO-Unit of Histology and Medical Embryology, Laboratory affiliated to Istituto Pasteur Italia-Fondazione Cenci Bolognetti, Sapienza University of Rome, Rome, Italy. [6]Department of Physiology and Pharmacology "V. Erspamer", Sapienza University of Rome, Rome, Italy. [7]Center for Human Technologies, Istituto Italiano di Tecnologia, Genoa, Italy. [8]Institute of Molecular Biology and Pathology, National Research Council, Rome, Italy. ✉E-mail: irene.bozzoni@uniroma1.it; julie.martone@cnr.it

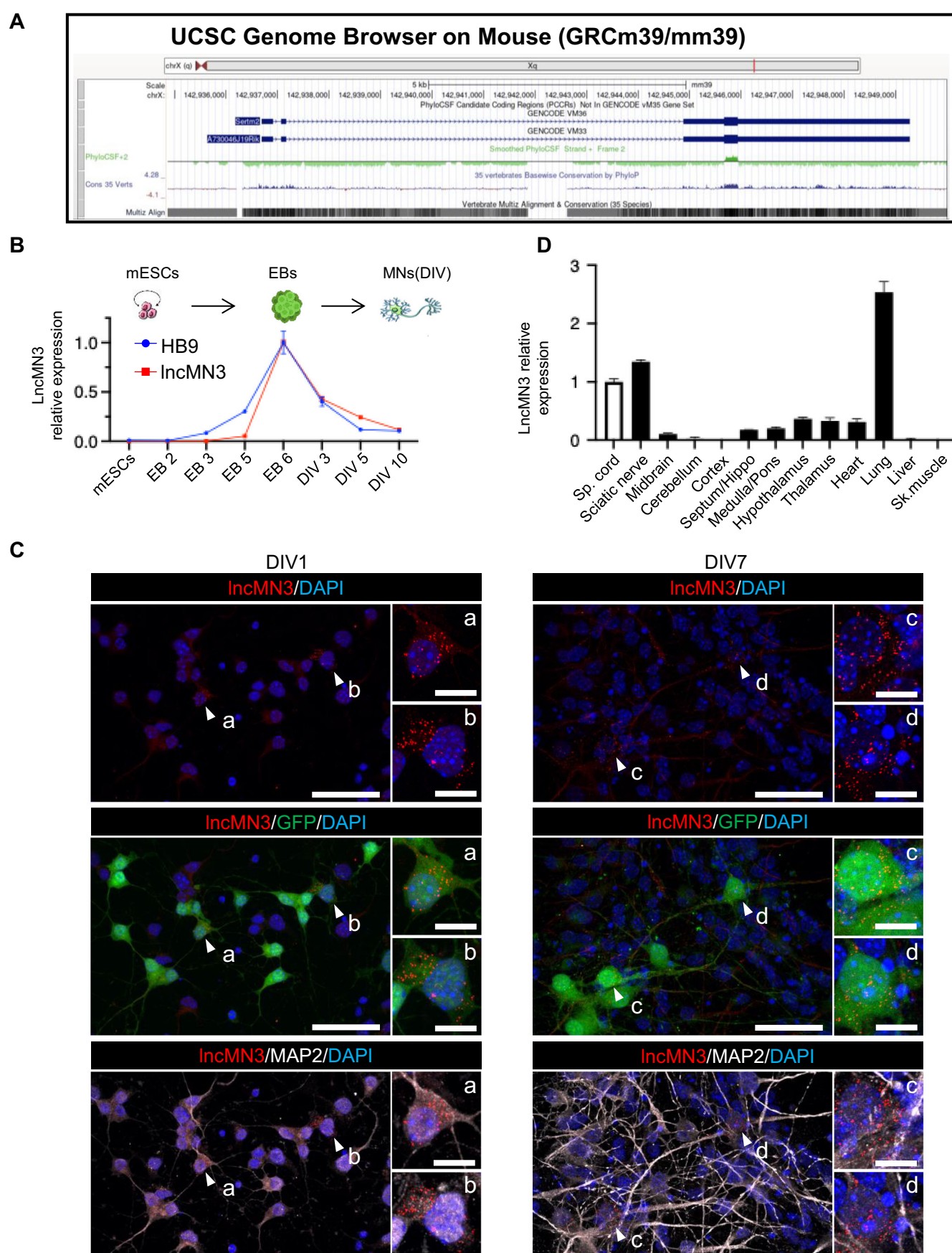

**Figure 1. Characterization of lncMN3 expression profile.**

(A) UCSC Genome Browser view showing the genomic organization of the murine A730046J19Rik/SERTM2 locus (GRCm39/mm39 assembly). The Smoothed PhyloCSF track (strand +, frame 2) is shown along with the PhyloP conservation among vertebrates. Transcript (Including UTRs): mm39 chrX:142,936,691-142,949,270 Size: 12,580 Total Exon Count: 3 Strand: +. Putative Coding Region: mm39 chrX:142,945,670-142,945,939 Size: 270. Coding Exon Count: 1. (B) qRT-PCR showing the expression levels of lncMN3 and HB9 transcripts during in vitro mESC differentiation to MNs at the indicated stages (EB embryoid bodies day n, DIV MNs days in vitro n). The expression levels were normalized against GAPDH mRNA and expressed as relative fold change with respect to EB day 6 samples set to a value of 1. The mean ± SEM is shown (n = 3, technical replicates). Representative experiment of three biological replicates. (C) Representative images showing FISH staining for lncMN3 RNA (red) combined with immunofluorescence for MAP2 neuron marker (white) and GFP (green) in motor neurons derived from HB9::GFP-mESCs 1 (DIV 1) and 7 days (DIV 7) after dissociation. Nuclei were marked with DAPI (blue). Larger panels: full-size images of confocal captions with a scale bar of 50 μm. White arrowheads indicate cells that are shown in magnified views. Smaller insets: Digitally magnified images of the indicated cells (refer to corresponding letters for identification), with a scale bar of 10 μm. (D) qRT-PCR showing the expression levels of lncMN3 from the indicated WT mouse tissues. The expression levels were normalized against the GAPDH mRNA and expressed as relative fold change with respect to the spinal cord sample set to a value of 1. The mean ± SEM is shown (n = 3, technical replicates). Source data are available online for this figure.

directly bind and modulate SERCA (S/ER Ca2+ ATPase), a pump that mediates Ca2+ uptake into the sarcoplasmic reticulum carrying out an important role in the regulation of cell calcium homeostasis. More in general several micropeptides involved in the regulation of the SERCA pump both in cardiac and skeletal muscle and in other non-muscle cells were identified, suggesting that small proteins could also be involved in the regulation of different membrane transporters (MacLennan et al, 2003; Anderson et al, 2016; Rathod et al, 2021). In accordance with this hypothesis, another previously annotated lncRNA (GM11549) was recently found to encode for a 63-aa long micropeptide called NEMEP that promotes glucose uptake by interacting with its transporters (GLUT1/GLUT3) to support the increased energy demand during mesoderm differentiation process (Fu et al, 2022).

An additional example that broadens the spectrum of micropeptides involved in the regulation of calcium homeostasis is represented by pTUNAR that is produced by lincTUNA bifunctional molecule (Tcl1 Upstream Neuron-Associated lincRNA, or megamind). Both the lncRNA and the micropeptide were identified in the central nervous system and in pancreatic β cells. The first role discovered was that of the lncRNA, which is predominantly nuclear and capable of forming complexes with RNA-binding proteins at the promoters of specific genes such as Nanog, Sox2, and Fgf4 to maintain pluripotency and promote neural lineage commitment (Lin et al, 2014). Only later, pTUNAR, the TUNA-encoded micropeptide, was identified and it was shown to be able to influence SERCA pump activity during neuronal differentiation and in pancreatic β cell (Senís et al, 2021; Li et al, 2021). Another study describes the role of two zebrafish misannotated lncRNAs able to encode for micropeptides in brain (linc-mipep and linc-wrb; Tornini et al, 2023). Tornini et al (2023) pointed out the essential role of linc-mipep and linc-wrb in regulating the development of vertebrate's specific brain cell types through the modulation of chromatin accessibility and gene expression. Finally, a recent study in differentiating neurons showed that a portion of the Malat1 lncRNA found in the cytoplasm, is translated into the M1 micropeptide (Xiao et al, 2024).

The last-mentioned research papers suggest that the functional role of micropeptides produced in the central nervous system may be greatly underestimated. In line with this observation, here we describe the function of SERTM2, a new identified micropeptide derived from a transcript previously annotated as noncoding RNA (Biscarini et al, 2018; Chuang et al, 2018). This 89 amino acids long micropeptide is specifically enriched in mouse motoneurons, conserved in humans, and characterized by a single transmembrane domain which localizes it at the level of plasma membrane.

We have shown that SERTM2 interacts with a two-pore domain K+ channel, TASK1, and that its depletion affects the resting membrane potential and excitability of mouse motor neurons.

## Results

### LncMN3 transcript is specifically enriched in spinal cord motor neurons

In a previous study, several lncRNAs enriched during in vitro mouse embryonic stem cells (mESCs) differentiation to motoneurons (MNs) were identified (Biscarini et al, 2018). In the paper cited above, MNs were FAC-sorted taking advantage of the presence of a GFP reporter under the control of the MN-specific HB9 promoter and the A730046J19Rik transcript was identified among the most enriched in the GFP+ population compared to the GFP-. The A730046J19Rik transcript, renamed lncMN3 in Biscarini et al, (2018), was originally annotated as a long intergenic noncoding RNA composed by three exons and located on the X chromosome. The same transcript was then annotated as coding gene (mm39 chrX:142,936,691–142,949,270; Fig. 1A) and was predicted to encode for a micropeptide called Serine-rich and transmembrane domain-containing protein 2 (SERTM2) that belongs to an uncharacterized family of proteins found in chordates. Proteins in this family are ~100 amino acids in length and according to Uniprot database their existence was only inferred from sequence homology (https://www.uniprot.org/uniprotkb/A0A1B0GWG4/entry).

We characterized the 3' end sequence of the lncMN3 transcript using rapid amplification of cDNA ends (RACE) and we confirmed the existence of the annotated isoform (Appendix Fig. S1A).

The analysis of the expression profile of lncMN3 during mESCs in vitro differentiation to MNs indicated that while it is not expressed in mESCs, it starts to be present in embryoid bodies at day 5 (EB5), reaches its maximum in EBs at day 6 (EB6) and decreases in the mixed population containing MNs (DIV3) obtained after cells dissociation (Fig. 1B). The observed decrease in expression is probably caused by a dilution effect due to the experimental protocol used for MN differentiation (Wichterle and Peljto, 2008) rather than to a real down-regulation. In fact, the MN population obtained upon EBs dissociation accounts for 40% of the mixed neural cell population; moreover, in contrast to the mixed population which continues to divide, MNs are postmitotic cells and their amount is diluted as differentiation proceeds (Capauto

et al, 2018). This interpretation was further supported by RNA fluorescence in situ hybridization (FISH) for lncMN3 performed on mESC-derived MNs one and seven days post-dissociation (DIV 1 and DIV 7 in Fig. 1C). Co-staining with HB9-GFP, shown in Fig. 1C, clearly indicated that, although the mixed population continued to proliferate during differentiation (GFP⁻ cells), the lncMN3 signal was mainly present in GFP⁺ cells. Such a correlation was further confirmed by qPCR data. As shown in Fig. 1B, the expression of lncMN3 closely paralleled that of HB9, reinforcing the specificity of lncMN3 for motor neuron population.

The enrichment of lncMN3 in MNs was also sustained by the analyses of existing single-cell RNA sequencing data derived from the same cellular system and differentiation protocol as described in our study (Carvelli et al, 2022; Data ref: Carvelli et al, 2021). The cluster identity assignment was performed as in Carvelli et al (2022) using the cell subpopulation markers described by Rizvi et al (2017). The following main cell populations were identified: neural precursors (NP), MN progenitors (MNP), early MN (EMN), late MN (LMN), and a class of neurons displaying interneuron (IN) markers (Fig. EV1A,B). As shown in Fig. EV1B, lncMN3 transcript was mainly detected in the late MN population.

Moreover, in 3-month-old mouse tissues, lncMN3 was specifically enriched in spinal cord, sciatic nerve and lung compared to other organs (Fig. 1D). The expression of lncMN3 transcript was also analyzed in the spinal cord at different stages of mouse development (E14, E16.5, P3, P7, P30, P60, and P90) revealing a constant expression in all samples examined with a peak in E16.5 and P3 (Fig. EV1C).

Finally, the enrichment of lncMN3 transcript in in vivo MNs was also confirmed by the interrogation of single-cell transcriptome atlases of developing mouse (Fig. EV1D, https://shiny.crick.ac.uk/scviewer/neuraltube/; Delile et at, 2019) and human spinal cord (Fig. EV1E, Rayon et al, 2021).

These data allow us to conclude that the lncMN3 transcript is enriched in MNs in both in vitro and in vivo model systems.

## Analysis of lncMN3 coding potential

The presence of short open reading frames (sORFs) is a feature shared by a substantial fraction of lncRNAs, however the existence and the biologically active role has been demonstrated only for a small proportion of peptides (Kondo et al, 2010) while the residual molecular RNA species remain annotated as noncoding. The initial A730046j19Rik/ lncMN3 noncoding annotation was determined by the fact that the murine transcript contains, according to ORF finder tool (https://www.ncbi.nlm.nih.gov/orffinder/), only short open reading frames (sORFs) that putatively encode for micropeptides, but their production was never demonstrated.

Despite the new annotation of A730046J19Rik/lncMN3 transcript as coding gene (SERTM2), when we assessed the coding potential of the murine transcript using two of the main available predictors, CPAT (Wang et al, 2013) and CPC (Kong et al, 2007), we retrieved a probability too low to confer a coding label to the transcript (CPAT prediction Appendix Fig. S1B; CPC prediction Appendix Fig. S1C). However, when we performed the same CPAT prediction on the human ortholog of lncMN3 (known as LINC00890 in Chuang et al, 2018 or CARDEL in Pereira et al, 2024) a positive coding capability was obtained (Appendix Fig. S1D).

We further analyzed the coding potential from an evolutionary point of view using Phylogenetic Codon Substitution Frequencies that considers not only the conservation of the coding sequence but also the high frequencies of synonymous codon and conservative amino acid substitutions as well as the low frequencies of other missense and nonsense substitutions (PhyloCSF; Mudge et al, 2019). The SERTM2 candidate coding region was identified in the lncMN3 noncoding transcript from both mouse (Fig. 1A) and human (Fig. 2A), and according to PhyloP track hub is highly conserved among vertebrates.

In addition, to predict translation initiation sites in the human and mouse lncMN3 nucleotide sequence, we used the TIS Predictor online tool, which identifies translation initiation codons with flanking nucleotides that closely match the Kozak sequence (https://www.tispredictor.com; Gleason et al, 2022). In both species, the ATG start codon of SERTM2 was identified and showed a good Kozak similarity score (see "position 1140" in mouse ENSMUST00000135687.2 sequence and "position 1081" in human ENST00000569275.2 sequence, Appendix Table S1).

Alignment analyses indicated that the aminoacidic identity between the predicted sequences of human and mouse SERTM2 micropeptides is almost 90% (Praline alignment, Fig. EV2A), and in both species according to InterPro prediction, three domains can be defined. An N-terminal non-cytoplasmic domain (from aa 1 to 34) a central transmembrane domain (from aa 35 to 58) and a cytoplasmic C-terminal domain (from aa 59 to 89 in mouse and 59 to 90 in human), suggesting that SERTM2 could be a type I transmembrane protein (Fig. 2A). Furthermore, a putative N-linked glycosylation site on the N-ter domain (Asn11) was also identified (https://www.uniprot.org/uniprotkb/A0A1B0GWG4/entry#ptm_processing).

Notably, SERTM2 was predicted to be a single-pass transmembrane protein also by Protter (Fig. EV2B; Omasits et al, 2014), deepTMHMM (Fig. EV2C; Hallgren, et al, 2022 bioRxiv), I-TASSER (Fig. EV2D, Yang et al, 2015) and Alphafold2 webservices (Fig. EV2E; Jumper et al, 2021).

We further investigated the lncMN3 translation hypothesis through sub-cellular fractionation of EB6 extract, which showed that the lncMN3 transcript was mainly localized in the cytoplasm (Fig. 2B and Biscarini et al, 2018), as also revealed by RNA-FISH for lncMN3 in mESC-derived MNs at 1 and 7 days of differentiation (Fig. 1C).

Moreover, polysome profiling analysis on EB6 extract showed the murine transcript associated with the heavy polysome fraction, with a pattern very similar to the coding control (ATP5O) (Fig. 2C). The association of lncMN3 with ribosomes was confirmed using the GWIPS-viz browser, which allows the visualization of ribo-seq and corresponding RNA-seq data from multiple studies in different species (Kiniry et al, 2018). Aggregated global tracks for mouse and human lncMN3 locus are shown in Fig. EV2F,G, respectively.

These features normally allow to discriminate efficiently translated transcripts from the untranslated ones (Gandin et al, 2014), adding further evidence for lncMN3 translation.

## SERTM2 is a type I transmembrane micropeptide

The coding potential of SERTM2-ORF was validated by western blot experiments on protein extracts obtained from HeLa cells transiently transfected with the mouse and human C-terminal

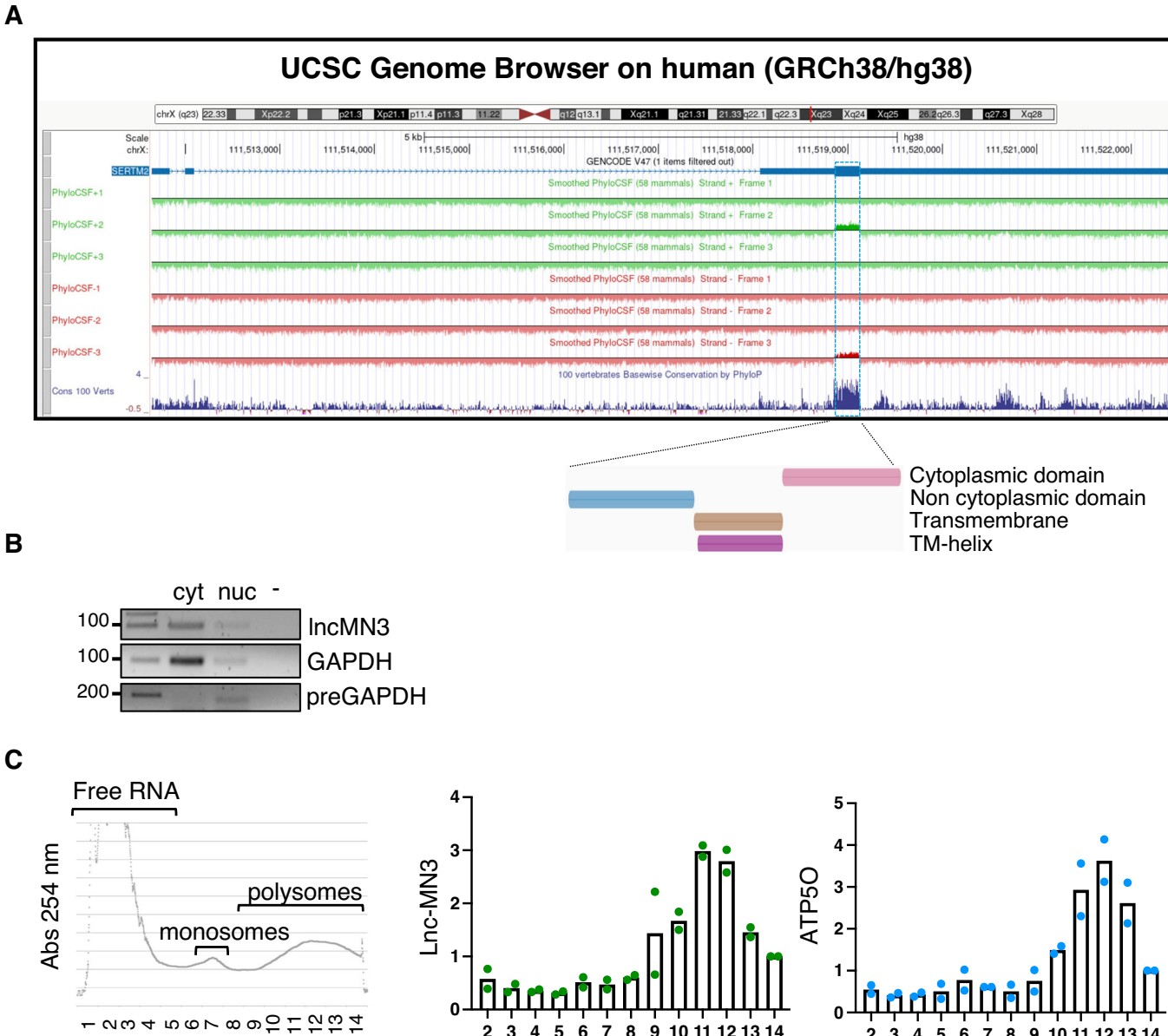

**Figure 2. lncMN3 coding potential.**

(A) UCSC Genome Browser view showing the genomic organization of the human SERTM2 locus (GRCh38/hg38 assembly). The Smoothed PhyloCSF track is shown along with the PhyloP conservation among vertebrates. The position of the sORF in the third exon, highlighted by the blue box, is indicated along with the domains predicted by interPro. (B) sqRT-PCR showing lncMN3 sub-cellular localization. RNA was isolated from EB day 6. Pre-GAPDH and GAPDH were used as nuclear and cytoplasmic controls respectively. Representative experiment of 3 biological replicates. (C) Ribosome profiling. EB day 6 lysate was subjected to sucrose gradient centrifugation to isolate fractions, including free RNA, monosomes, and polysomes. RNAs were then extracted from these fractions, and the levels of LncMN3 were quantified by qPCR. ATP5O was used as coding control. Data are expressed as relative fold change with respect to fraction 14 set to a value of 1. The mean of two biological replicates (dots) is shown. Source data are available online for this figure.

flagged SERTM2-ORF overexpressing constructs (mSERTM2-FLAG and hSERTM2-FLAG respectively) or an empty plasmid (pcDNA, Fig. 3A). The predicted plasma membrane localization was confirmed by sub-cellular fractionation (Fig. 3B) and immunofluorescence analysis using anti-FLAG antibodies (Fig. 3C) exploiting the described HeLa cell system. Afterward, a CRISPR/CAS9 genome editing strategy was employed to produce mESCs FLAG KI clones inserting a FLAG sequence downstream the sORF

(Fig. EV3A). Western blot analyses on FLAG KI EB6 protein extract confirmed the presence of the micropeptide in the membrane fraction (Fig. 3D). A custom rabbit monoclonal antibody against the conserved C-terminal cytoplasmic domain of SERTM2 (72:89 residues, Fig. EV2B) was then produced (Thermo-Fisher) and tested by performing western blot and immunofluor-escence analyses on HeLa cells transfected with the plasmid overexpressing the mSERTM2-FLAG protein or an empty vector

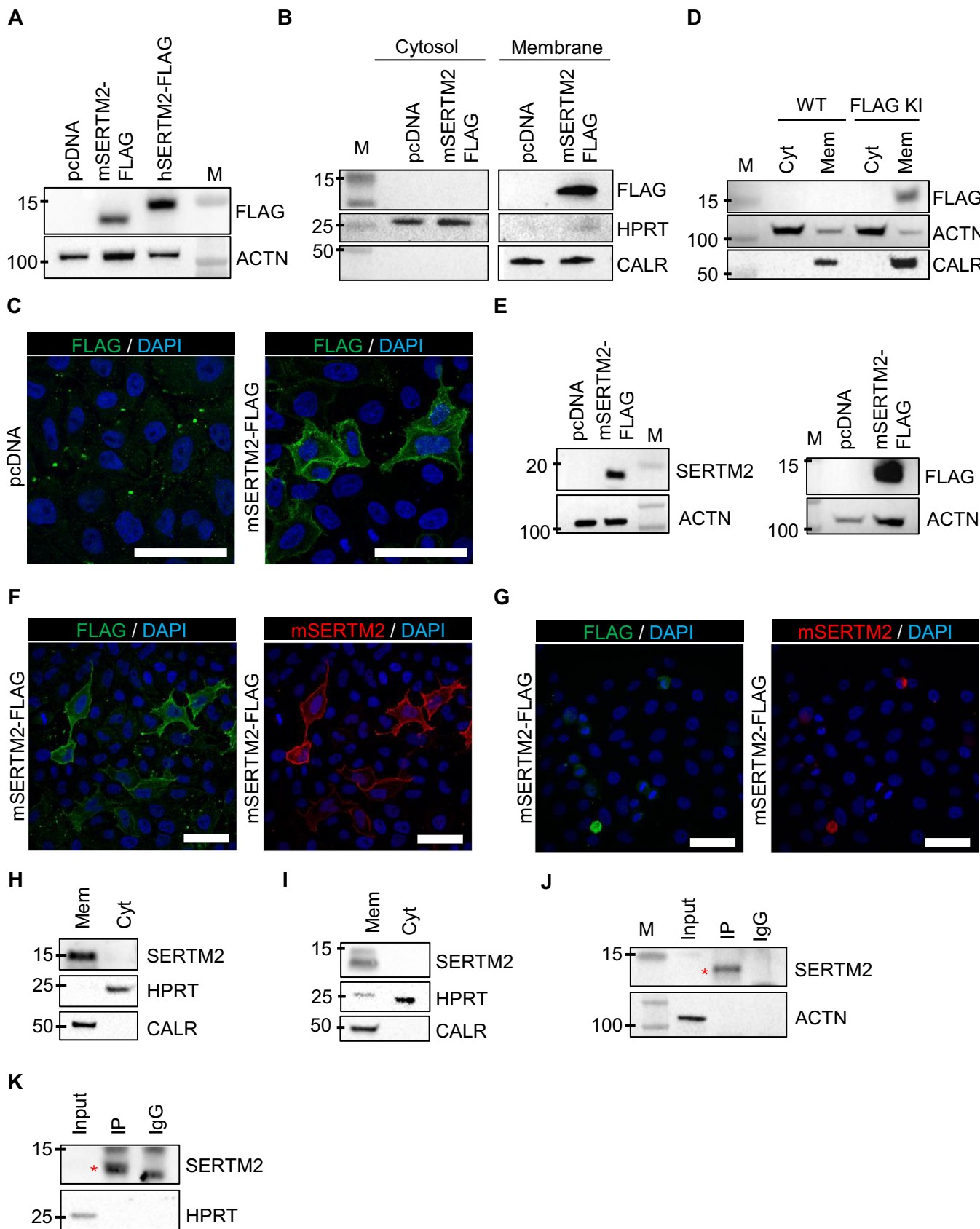

**Figure 3. SERTM2 micropeptide expression.**

(A) Western blot showing the expression of SERTM2 from HeLa cells protein extract overexpressing murine (mSERTM2) or human (hSERTM2) SERTM2-FLAG protein using anti-FLAG antibodies (FLAG). Actinin (ACTN) was used as a loading control. Representative experiment of three biological replicates. (B) Western blot on isolated membrane and cytosolic protein extracts using anti-FLAG antibodies (FLAG). HPRT and Calreticulin (CALR) were used as cytosolic and membrane controls respectively. Representative experiment of three biological replicates. (C) Immunofluorescence for FLAG (green signal) in combination with DAPI staining (blue) performed on HeLa cells transfected with an empty plasmid (left, pcDNA) and overexpressing SERTM2-FLAG (right). Scale bar corresponding to 50 µm. Representative experiment of three biological replicates. (D) Western blot on isolated membrane and cytosolic protein extracts from Flag KI mESCs differentiated to EBs day 6 using anti-FLAG antibodies (FLAG). ACTN and CALR antibodies were used as cytoplasmic and membrane controls respectively. Representative experiment of three biological replicates. (E) Western blot showing the expression of SERTM2 from HeLa cells protein extract overexpressing murine SERTM2-FLAG. Left panel: custom antibodies against SERTM2 were used, the marker lane (M) is shared with Fig. 4A. Right panel: antibodies against FLAG were used. ACTN was used as loading control. Representative experiment of three biological replicates. (F) Immunofluorescence for FLAG (green signal, left panel) or SERTM2 (red signal, right panel) detection in combination with DAPI staining (in blue) performed on HeLa cells transfected with a plasmid overexpressing mSERMT2-FLAG. Samples were subjected to a permeabilization treatment (Triton 0.1%). Scale bar corresponding to 50 µm. Representative experiment of three biological replicates. (G) Immunofluorescence for FLAG (green signal, left panel) or SERTM2 (red signal, right panel) detection in combination with DAPI staining (in blue) performed on HeLa cells transfected with a plasmid overexpressing mSERMT2-FLAG. The permeabilization treatment was omitted. Scale bar corresponding to 50 µm. Representative experiment of three biological replicates. (H) Western blot on isolated membrane and cytosolic protein extracts from WT EB day 6 using anti-SERTM2 antibodies. HPRT and CALR antibodies were used as cytoplasmic and membrane controls respectively. Representative experiment of three biological replicates. (I) WB analyses on isolated membrane and cytosolic protein extracts from hiPSCs 5 days after the induction of motor neuronal differentiation. HPRT and CALR antibodies were used as cytoplasmic and membrane controls respectively. Representative experiment of three biological replicates. (J) Western blot with anti-SERTM2 antibodies on proteins obtained after SERTM2 immunoprecipitation from 3-month-old mouse spinal cord extract (P90). The red asterisk corresponds to a specific SERTM2 band. Input sample accounts for 1.5% of the extract. ACTN was used as a negative control. Representative experiment of three biological replicates. (K) Western blot with anti-SERTM2 antibodies on proteins obtained after SERTM2 immunoprecipitation from E16.5 embryonal spinal cord extract. The red asterisk corresponds to specific SERTM2 band. Input sample accounts for 1.5% of the extract. HPRT was used as a negative control. Representative experiment of three biological replicates. Source data are available online for this figure.

(pcDNA). As shown in Fig. 3, the custom antibody revealed the same expression pattern obtained with anti-FLAG Ab both in western blot experiments (Fig. 3E) and immunofluorescence analyses (Fig. 3F).

To assess that SERTM2 is a type I transmembrane protein, according to the predicted topology, immunofluorescence staining was performed on HeLa cells transfected with the mSERTM2-FLAG overexpressing plasmid. When cells were treated with Triton (0.1%) to permeabilized the cell membrane and make accessible intracellular antigens, the fluorescence signal was detected using antibodies against both FLAG and SERTM2 which recognize the C-terminal part of the micropeptide (Fig. 3F). However, when Triton treatment on transfected HeLa cells was omitted, fluorescence was not detected with neither antibodies (Fig. 3G), confirming that SERTM2 is a type I transmembrane protein (Chou and Elrod, 1999). HeLa cells transfected with the empty vector (pCDNA) and treated or not with Triton were used as negative controls (Fig. EV3B,C).

The endogenous peptide was detected by western blot analyses on membrane protein extract of EB6 derived from mouse ESCs (Fig. 3H); this data allowed us to also confirm the membrane localization of the endogenous SERTM2 by biochemical fractionation. Moreover, the existence of the human SERTM2 micropeptide was validated by western blot analyses on membrane protein extracts from iPSC-derived MNs (Fig. 3I).

Concerning mouse tissues, several buffers were tested for the extraction of total and membrane proteins from spinal cord homogenates, but a specific micropeptide signal could never be detected (Fig. EV3D). This was probably because the spinal cord is a complex tissue composed of different cell types of which only a small fraction are motor neurons, the cells in which SERTM2 is expressed. Therefore, we enriched SERTM2 by immunoprecipitating the extract with SERTM2 antibody before western blot analysis. With this procedure, a specific signal corresponding to SERTM2 was identified in the immunoprecipitated spinal cord protein extracts obtained from 3-month-old mice (P90—Fig. 3J) and embryos (E16.5—Fig. 3K). The same immunoprecipitation

experiment was performed, as negative control, on liver extracts (P90) where according to real-time PCR analysis lncMN3 transcript is not expressed (Fig. 1C) and a SERTM2 specific signal was not detected (Fig. EV3E). Surprisingly, even if lncMN3 transcript was expressed in the lung (Fig. 1C), the WB performed on SERTM2-immunoprecipitated lung protein extract did not retrieve any signal (Fig. EV3F). This experiment suggests that in this tissue the lncMN3 transcript could have a function as long noncoding RNA, independently from the micropeptide production.

These data allowed us to conclude that, at least in the spinal cord, lncMN3 encodes for the SERTM2 transmembrane micropeptide.

## SERTM2 loss of function results in altered membrane potential and neuron excitability

Two different CRISPR/CAS9 genome editing approaches were used in mESCs to study the function of the lncMN3 transcript. The first one was aimed at abolishing the entire lncMN3 transcript by inserting a poly-A site downstream the lncMN3 second exon (lncMN3-KO; Fig. EV4A), while the second one at inhibiting the micropeptide production through a frameshift mutation in the uOFR, thus preserving the overall structure of the locus (ΔSERTM2; Fig. EV4B). As shown in Fig. EV4C (lower panel), the expression of lncMN3 was abolished in the lncMN3-KO selected clone, while the level of the transcript was unchanged in the lncMN3-ΔORF selected clone. The presence of the nucleotide deletion in the lncMN3-ΔORF was verified by PCR amplification and sequencing (Fig. EV4D). The absence of the micropeptide production in lncMN3-KO and lncMN3-ΔORF clones was confirmed by western blot analyses using antibodies against SERTM2 (Fig. 4A).

To assess the possible function of SERTM2 in motor neuron physiology, patch-clamp experiments were performed in WT, lncMN3-KO, and lncMN3-ΔORF motor neurons differentiated from mESC at 7, 14, and 21 days of maturation (DIV 7, DIV14, and DIV21). Initially evoked action potentials (APs) were recorded (Fig. 4B). The relationship between injected current and the number of

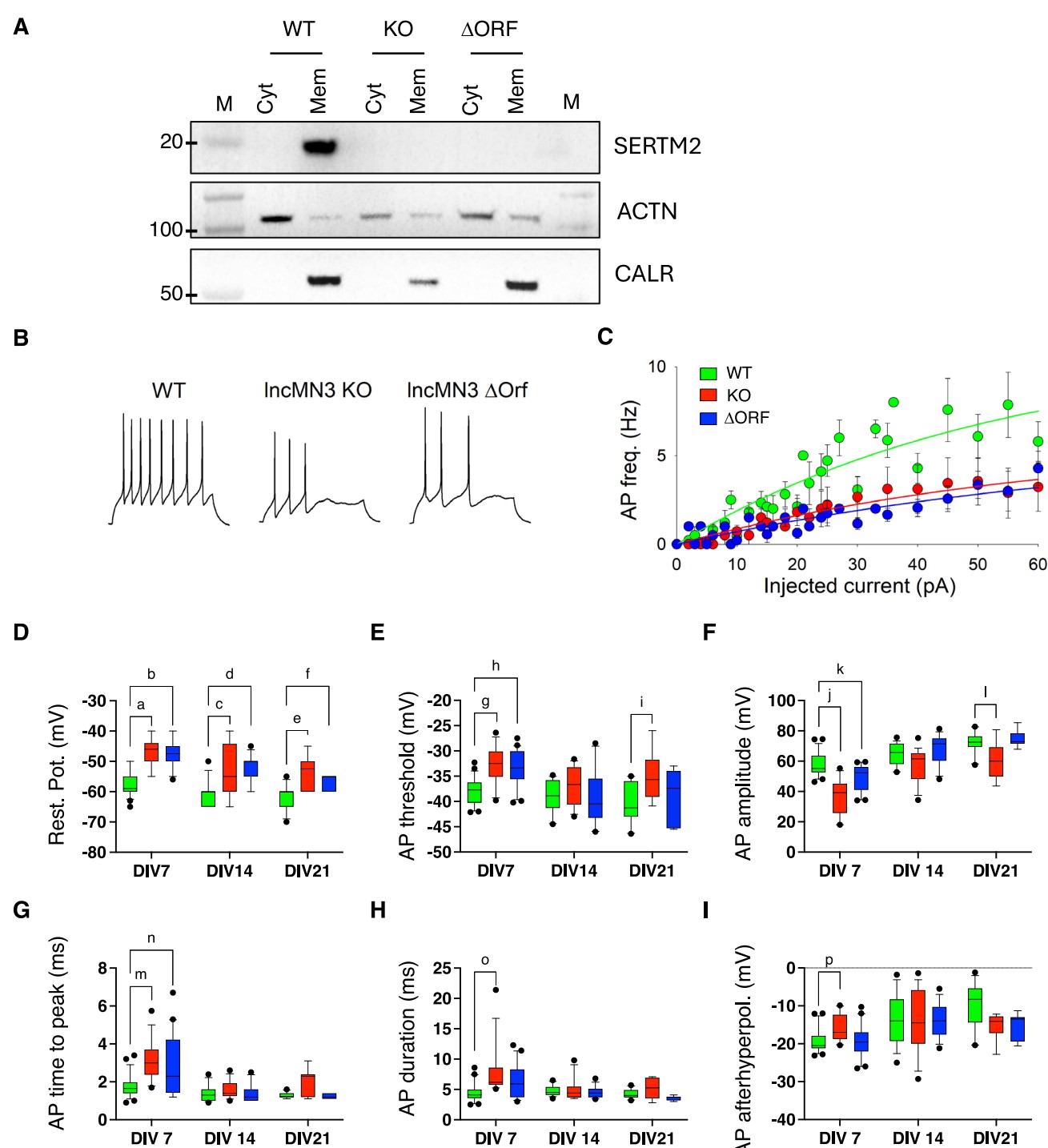

elicited APs showed a lower excitability of lncMN3-KO and lncMN3-ΔORF neurons in comparison to WT cells, at DiV 7 (Fig. 4C). The resting potential of lncMN3-KO and lncMN3-ΔORF neurons was significantly depolarized compared to WT neurons, at each developmental stage (Fig. 4D). Furthermore, the analyses of other AP properties (Fig. 4E–I) showed alteration in lncMN3-KO and lncMN3-ΔORF neurons at DIV 7: the AP threshold was depolarized (Fig. 4E), with reduced AP amplitude (Fig. 4F) and

longer AP time to peak (Fig. 4G). Total duration and after-hyperpolarization were significantly altered only in lncMN3-KO (Fig. 4H,I). These data indicate that mutant neurons show a lower excitability than the WT neurons.

However, spontaneous synaptic transmission among mESC-derived neurons (Fig. EV4E,F) was not affected by the absence of SERTM2: the frequency of spontaneous synaptic currents (Fig. EV4G) and spontaneous action potentials (Fig. EV4H) did

◄

**Figure 4. Block of SERTM2 expression alters the functional properties of action potentials of mESC-derived motor neuron.**

(A) Western blot on isolated membrane and cytosolic protein extracts from WT, KO and ΔORF EBs day 6 with SERTM2 antibody. ACTN and CALR antibodies were used as cytoplasmic and membrane controls respectively. The marker lane (M) is shared with Fig. 3E left panel. Representative experiment of 3 biological replicates. (B) Typical traces of evoked action potentials recorded in current-clamp configuration from a WT (left), a lncMN3-KO (middle), and a lncMN3-ΔORF neuron at DIV 7 (injected current 40 pA, 1 s). (C) Input–output relationship for WT (green), lncMN3-KO (red) and lncMN3-ΔORF (blue) neurons, at DIV 7. Please note the higher excitability for WT neurons. $n = 24$, $n = 13$, $n = 20$ for WT, lncMN3-KO and lncMN3-ΔORF, respectively. Error bars correspond to standard deviations. (D) Box and whisker plots representing resting potential of WT, lncMN3-KO and lncMN3-ΔORF neurons, at DIV 7 ($n = 24$, $n = 13$, $n = 20$ for WT, lncMN3-KO and lncMN3-ΔORF, respectively), 14 ($n = 15$, $n = 18$, $n = 13$ for WT, lncMN3-KO and lncMN3-ΔORF, respectively) and 21 ($n = 11$, $n = 8$, $n = 7$ for WT, lncMN3-KO and lncMN3-ΔORF, respectively), as indicated. The plots display the 90th and 10th percentiles at the whiskers, the 75th and 25th percentiles at the boxes, and the median at the central line. Black circles represent outlier data outside the 10th and 90th percentiles. a, $P = 2.02\text{E-}05$; b, $P = 1.1\text{E-}6$; c, $P = 4.6\text{E-}3$; d, $P = 1.3\text{E-}3$, e, $P = 4.8\text{E-}3$, f, $P = 4.2\text{E-}2$. Statistical analyses were performed using the Kruskal–Wallis test followed by Dunn's Multiple Comparison test. (E) Box and whisker plots representing the AP threshold of WT, lncMN3-KO, and lncMN3-ORF neurons, at DIV 7 ($n = 24$, $n = 13$, $n = 20$ for WT, lncMN3-KO, and lncMN3-ΔORF, respectively), 14 ($n = 15$, $n = 18$, $n = 13$ for WT, lncMN3-KO, and lncMN3-ΔORF, respectively) and 21 ($n = 11$, $n = 8$, $n = 7$ for WT, lncMN3-KO, and lncMN3-ΔORF, respectively), as indicated. The plots display the 90th and 10th percentiles at the whiskers, the 75th and 25th percentiles at the boxes, and the median at the central line. Black circles represent outlier data outside the 10th and 90th percentiles. g, $P = 5\text{E-}5$; h, $P = 5\text{E-}5$; i, $P = 3.9\text{E-}2$. Statistical analyses were performed using one-way ANOVA followed by Holm–Sidak's Multiple Comparison test. (F) Box and whisker plots representing the AP amplitude of WT, lncMN3-KO, and lncMN3-ΔORF neurons, at DIV 7 ($n = 24$, $n = 13$, $n = 20$ for WT, lncMN3-KO, and lncMN3-ΔORF, respectively), 14 ($n = 15$, $n = 18$, $n = 13$ for WT, lncMN3-KO, and lncMN3-ΔORF, respectively) and 21 ($n = 11$, $n = 8$, $n = 7$ for WT, lncMN3-KO, and lncMN3-ΔORF, respectively), as indicated. The plots display the 90th and 10th percentiles at the whiskers, the 75th and 25th percentiles at the boxes, and the median at the central line. Black circles represent outlier data outside the 10th and 90th percentiles. j, $P = 2.4\text{E-}8$; k, $P = 3.5\text{E-}3$; l, $P = 1.8\text{E-}2$. Statistical analyses were performed using one-way ANOVA followed by Holm–Sidak's Multiple Comparison test. (G) Box and whisker plots representing the AP time to peak of WT, lncMN3-KO, and lncMN3-ΔORF neurons, at DIV 7 ($n = 24$, $n = 13$, $n = 20$ for WT, lncMN3-KO, and lncMN3-ORF, respectively), 14 ($n = 15$, $n = 18$, $n = 13$ for WT, lncMN3-KO, and lncMN3-ΔORF, respectively) and 21 ($n = 11$, $n = 8$, $n = 7$ for WT, lncMN3-KO, and lncMN3-ΔORF, respectively), as indicated. The plots display the 90th and 10th percentiles at the whiskers, the 75th and 25th percentiles at the boxes, and the median at the central line. Black circles represent outlier data outside the 10th and 90th percentiles. m, $P = 6.6\text{E-}4$; n, $P = 3.4\text{E-}2$. Statistical analyses were performed using the Kruskal–Wallis test followed by Dunn's Multiple Comparison test. (H) Box and whisker plots representing the AP duration of WT, lncMN3-KO, and lncMN3-ΔORF neurons, at DIV 7 ($n = 24$, $n = 13$, $n = 20$ for WT, lncMN3-KO, and lncMN3-ΔORF, respectively), 14 ($n = 15$, $n = 18$, $n = 13$ for WT, lncMN3-KO and lncMN3-ΔORF, respectively) and 21 ($n = 11$, $n = 8$, $n = 7$ for WT, lncMN3-KO, and lncMN3-ΔORF, respectively), as indicated. The plots display the 90th and 10th percentiles at the whiskers, the 75th and 25th percentiles at the boxes, and the median at the central line. Black circles represent outlier data outside the 10th and 90th percentiles. o, $P = 7.1\text{E-}4$. Statistical analyses were performed using the Kruskal–Wallis test followed by Dunn's Multiple Comparison test. (I) Box and whisker plots representing the AP afterhyperpolarization of WT, lncMN3-KO, and lncMN3-ΔORF neurons, at DIV 7 ($n = 24$, $n = 13$, $n = 20$ for WT, lncMN3-KO and lncMN3-ΔORF, respectively), 14 ($n = 15$, $n = 18$, $n = 13$ for WT, lncMN3-KO, and lncMN3-ΔORF, respectively) and 21 ($n = 11$, $n = 8$, $n = 7$ for WT, lncMN3-KO, and lncMN3-ΔORF, respectively), as indicated. The box plots display the 90th and 10th percentiles at the whiskers, the 75th and 25th percentiles at the boxes, and the median at the central line. Black circles represent outlier data outside the 10th and 90th percentiles. p, $P = 5.3\text{E-}3$. Statistical analyses were performed using the Kruskal–Wallis test followed by Dunn's Multiple Comparison test. Source data are available online for this figure.

not show any difference between WT, lncMN3-KO and lncMN3-ΔORF neurons.

Interestingly, lncMN3-KO and lncMN3-ΔORF genotypes produced very similar phenotypes, suggesting that the observed functional alterations are due to the absence of the micropeptide rather than the lncMN3 transcript that, in ΔORF clones, is expressed at WT levels.

## SERTM2 interacts with the two-pore domain potassium channel TASK1

In accordance with patch-clamp results, we hypothesized that SERTM2 could interact with an ion channel to influence the membrane resting potential. According to literature data, the main channels involved in establishing the resting potential in neurons belong to the family of Two-pore domain (K2P) "leak" K(+) channels (Lesage and Lazdunski, 2000; Berg et al, 2004). Among them, TASK1 (KCNK3) and TASK3 (KCNK9) appear to be the main determinants of this parameter in motor neurons (Talley et al, 2001; Karschin et al, 2001; Berg et al, 2004). Interestingly, these channels are included in the list of the first 1000 transcripts co-expressed with SERTM2 in EB6 based on the data obtained from the mentioned single-cell analysis (Appendix Table S2). These findings suggested that the observed resting potential modulation could be mediated by an interaction between SERTM2 and TASK1/3 channels. Even if we cannot exclude an interaction with TASK3, the experiments were focused on TASK1 since the X-ray crystal structure of TASK3 was not available. To gain additional structural

information into the interaction between SERTM2 and TASK1, the complex between the two proteins was predicted utilizing the state-of-the-art AlphaFold3 (AF3) algorithm (Fig. 5A; Abramson et al, 2024). Part of the transmembrane helix (residues 36–48) of SERTM2 is precisely docked onto the closed conformation of the two-pore domain potassium ion channel (PDB: 6RV3; Rödström et al, 2020), revealing regions of potential interaction with helices 101–116, 158–180, and 184–200 of TASK1. Subsequently, for each predicted solution of AF3, the DALI server (Holm, 2022) was employed to discern potential similar complexes within the Protein Data Bank (PDB). Through this iterative process, a remarkable discovery of structural similarity emerged, revealing an unexpected parallel between the SERTM2/TASK1 interaction and the established cryo-EM complex of the voltage-gated potassium channel KCNQ3 and its regulator KCNE3 (PDB: 6V00; Sun et al, 2020; Fig. 5B). The juxtaposition of the cryo-EM structure of KCNQ3/KCNE3 and the model of SERTM2 bound to TASK1 (center panel of Fig. 5B) underscores striking similarities in their conformations, suggesting a process of convergent evolution and a conserved mechanism underlying the modulation of potassium ion channels.

The predicted interaction between SERTM2 and TASK1 was tested by means of co-immunoprecipitation experiments performed in HeLa cells overexpressing SERTM2 and TASK1-FLAG. The enrichment of SERTM2 and TASK1 in the SERTM2-immunoprecipitated fraction, was assessed by western blot analysis using anti-SERTM2 and anti-FLAG antibodies. The results shown in Fig. 5C confirm that following the IP for SERTM2, an enrichment of TASK1 in the IP fraction was observed indicating

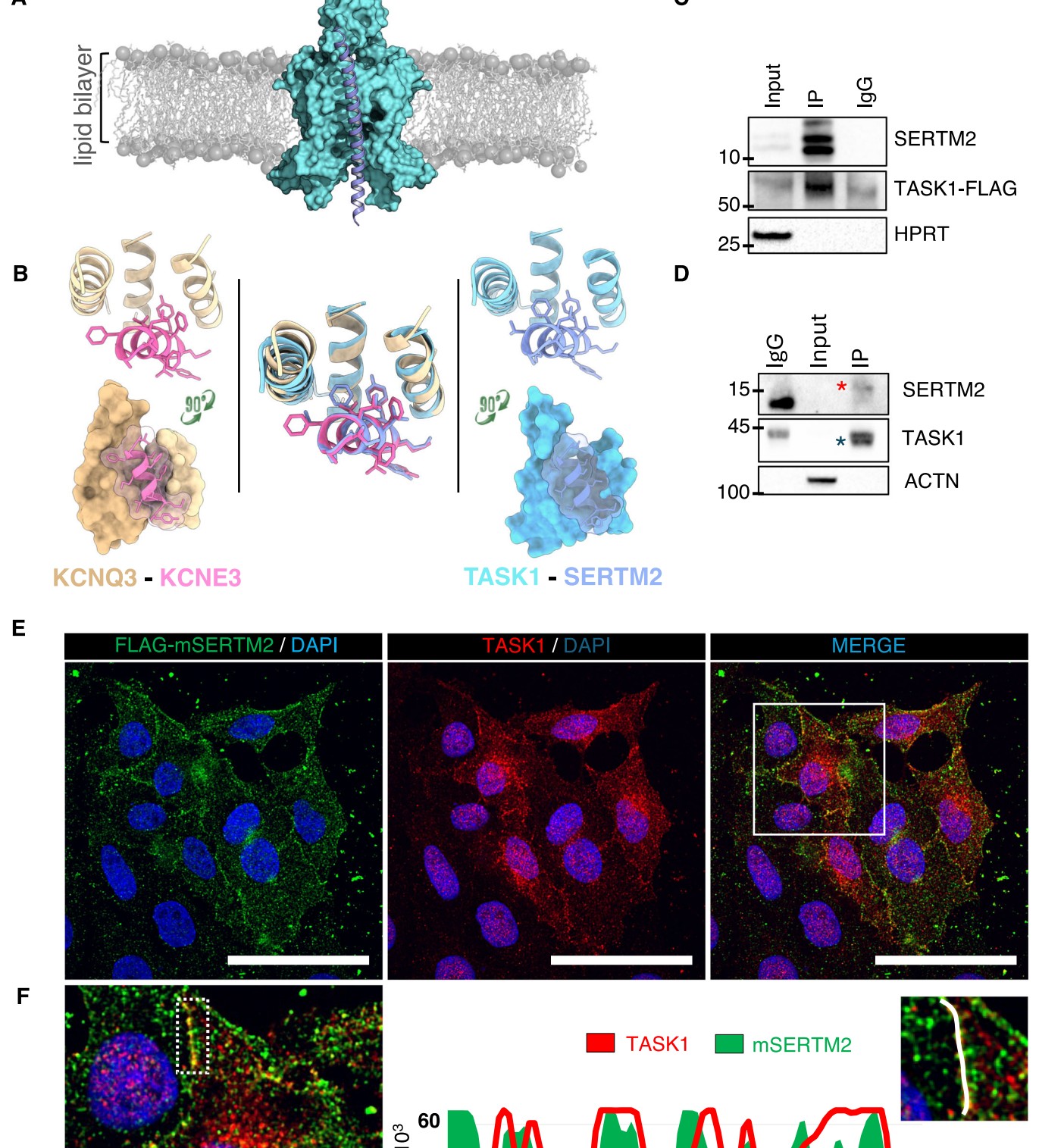

◀  **Figure 5.  SERTM2 interacts with the ion channel TASK1.**

(A) Representation with the transparent surface of lipid bilayer, TASK1 chains A and B in cyan surface (reference PDB: 6RV3), and the transmembrane helix of SERTM2 (22–65) in slate cartoons. The lipid bilayer is shown as reference and its relative position compared to the complex predicted with the OPM server (https://opm.phar.umich.edu/proteins/4644; Lomize et al, 2011). (B) (Left): Cryo-EM structure of KCNE3 (magenta, helix 58–70) in complex with KCNQ3 (pale brown, helices 260–285, 297–311, 320–336) (PDB: 6V00). (Right) 3D-model of SERTM2 (slate cartoon) helix 36–48 docked to the crystal structure of the human two-pore domain potassium ion channel TASK1 in closed conformation (PDB: 6RV3). Helices 101–116, 158–180, and 184–200 (cyan) are shown as cartoon (upper panel) and surface (lower panel). (Center) Superposition of upper left and right panels. (C) Western blot on protein lysates from HeLa cells transfected with TASK1-FLAG and SERTM2 overexpressing plasmids. Protein extracts were immunoprecipitated with anti-SERTM2 antibody followed by immunoblotting with anti-SERTM2 and anti-FLAG antibodies. Input sample accounts for 1.5% of the extract. HPRT was used as a negative control. Representative experiment of three biological replicates. (D) Western blot on proteins from 3-month-old mouse spinal cord extract (P90). Protein extracts were immunoprecipitated with anti-SERTM2 antibody followed by immunoblotting with anti-SERTM2 and anti-TASK1 antibodies. Red and blue asterisks correspond respectively to specific SERTM2 and TASK1 band. Input sample accounts for 1.5% of the extract. ACTN was used as a negative control. Representative experiment of 2 biological replicates. (E) Immunofluorescence for FLAG-SERTM2 (green signal) and TASK1 (red signal) performed on HeLa cells overexpressing SERTM2-FLAG and TASK1 respectively (left and middle panel). Composite image of FLAG-SERTM2/TASK1 is shown in right panel. Scale bars correspond to 50 μm. Representative experiment of three biological replicates. (F) Digital magnification of region depicted in (E) and Fluorescence Intensity (FI) profile of FLAG-SERTM2/TASK1 signals in the insert highlighted by dashed square. Pearson's correlation coefficients (PCC) indicate the colocalization index between FLAG-SERTM2 and TASK1 fluorescence scanned along the solid white line shown in the insert. Scale bars correspond to 10 μm. Source data are available online for this figure.

that the two proteins form a complex in vitro, at least in conditions of overexpression.

To verify that the interaction between SERTM2 and TASK1 was occurring also in physiological conditions, Co-IP experiments were performed on mouse spinal cord protein extracts using anti-SERTM2 antibodies. As shown in Fig. 5D also in this case we observed enrichment of TASK1 in the IP fraction. To further validate this association, immunofluorescence analyses on HeLa cells overexpressing SERTM2-FLAG and TASK1 proteins ware performed. As shown in Fig. 5E,F, FLAG signal co-localizes with TASK1 at the plasma membrane. Moreover, linescan analysis combined with Pearson's correlation quantification (Fig. 5F, right) support the observation that SERTM2 and TASK1 ion channel are spatially connected in the same membrane-associated complex.

## SERTM2-KO mouse model

To characterize the in vivo function of SERTM2 micropeptide, a knockout mouse model was produced by CRISPR-Cas9 genome editing approach using a single-strand DNA oligonucleotide as donor sequence. The result of the genome editing was the deletion of the entire sORF sequence except for the first 11 nucleotides that were mutated to insert a stop codon in frame with a disrupted ATG codon (Fig. EV5A). Four hemizygous males were obtained through zygotic injection, and their breding with WT females produced heterozygous females; homozygous females were obtained by F2 animals breeding. The expected editing was confirmed by genomic PCR amplification and sequencing (Fig. EV5B). As shown by RT-PCR analyses, the expression levels of lncMN3 transcript in the spinal cord of KO mice were unchanged when compared to WT one (Fig. 6A). However, a reduction in the PCR amplicon size was observed in accordance with sORF deletion (Fig. 6B). The absence of SERTM2 micropeptide production in hemizygous male was verified by western blot analyses on spinal cord extract immunoprecipitated with SERTM2 antibodies (Fig. 6C). SERTM2 hemizygous and homozygous KO mice were born in expected mendelian ratios and were normally viable after birth (Appendix Table S3). No significant differences were found between genotypes in body weight (Fig. EV5C). To investigate the neuromuscular capacity of SERTM2-KO mice in a condition that requires high neuronal excitability, a treadmill exhaustion test was used.

The criteria for exhaustion are defined as spending five consecutive seconds lying on the shock grid and failing to resume

running despite repeated aversive stimuli. The distance traveled and the number of shocks were recorded for each of the five running sessions. The first parameter remained constant in the different groups of 3- and 9-month-old mice (Hem, Homo, and WT) suggesting that the locomotor capacity is not impaired by the absence of SERTM2 (Fig. EV5D,E). On the other hand, the number of shocks resulted to be increased in hemizygous male and homozygous female when compared to their respective WT-aged and sex-matched controls, and this increase was exacerbated with subsequent runs (Fig. 6D,E). The increase in shock number remained constant with increasing age as shown by the analyses of hemizygous 9-month-old males compared to WT ones (Fig. EV5F,G), excluding the involvement of a degenerative effect.

The results obtained from locomotor test suggest that SERTM2-KO mice do not present locomotor impairment nor obvious phenotypes other than an increase in the number of shocks received during treadmill, that could be explained by the lower excitability of lncMN3-ΔORF and KO-MNs observed in vitro.

## Discussion

Human and mouse genomes consist of nineteen and twenty-one thousand protein-coding genes, respectively (Frankish et al, 2023). These numbers were determined using algorithms that, among other parameters, have a cutoff of 100 codons. However, since the annotation of the first murine lncRNA-derived micropeptides (Anderson et al, 2015), it became evident that this rule was too stringent, resulting in the loss of several proteins or in their misannotation as lncRNAs. Furthermore, ribosome profiling sequencing data indicated that ~40% of lncRNAs in humans and 80% in mice were associated with ribosomes (Ingolia et al, 2011; Guo et al, 2010; Ruiz-Orera et al, 2014). These data suggest that, although not extensively described in literature, a significant number of lncRNA-derived micropeptides may exist in the central nervous system, where 40% of all long noncoding RNAs identified in the human genome are specifically expressed (Derrien et al, 2012; Ulitsky et al, 2011). LncMN3 could fall either into the category of misannotated lncRNAs or into that of dual-function lncRNAs. In this study, we demonstrate the lncMN3 translational competence and delineate the function of the resulting SERTM2 micropeptide in motor neurons. Regarding the initial aspect,

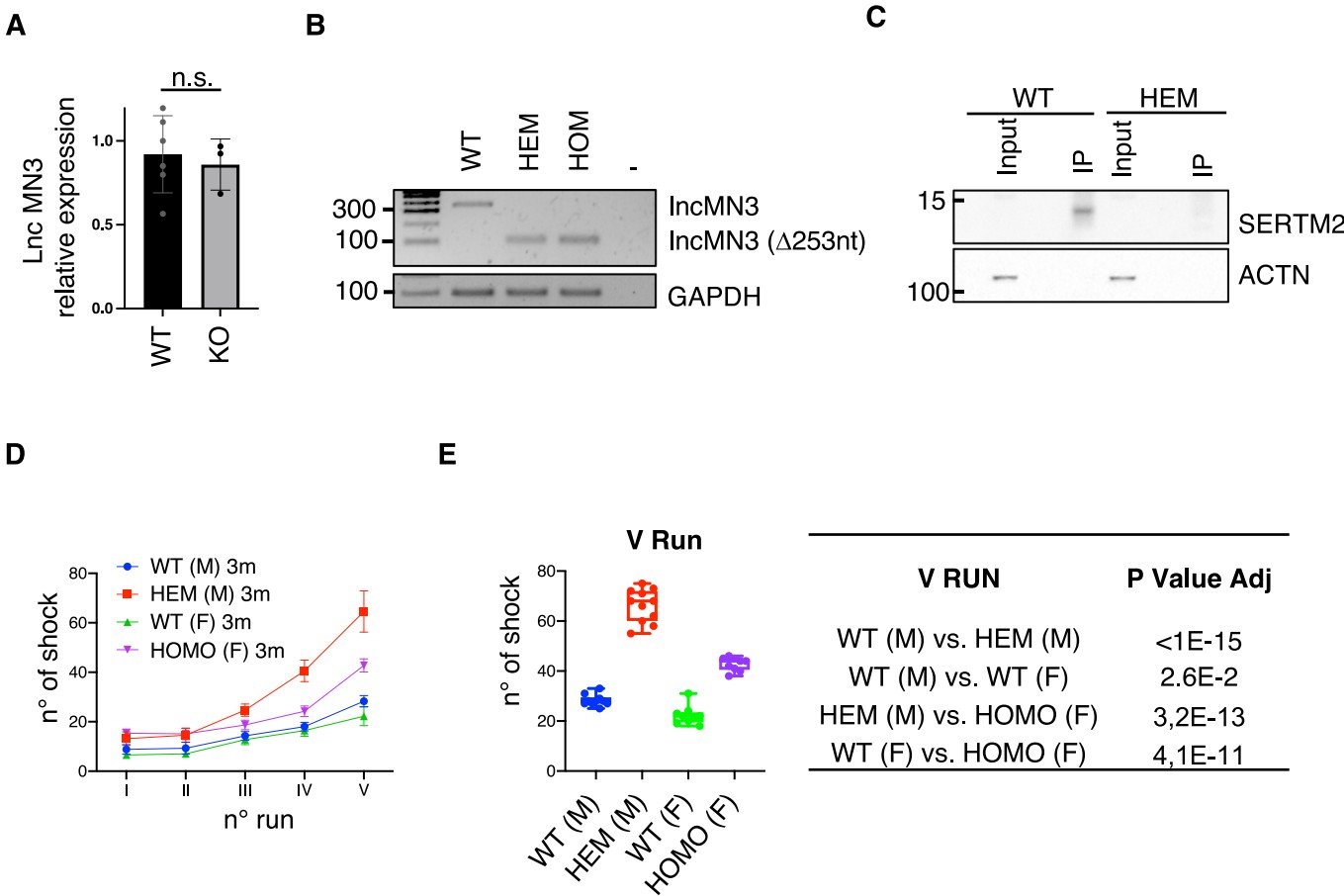

**Figure 6.  SERTM2-KO mouse model.**

(A) qRT-PCR showing the expression levels of lncMN3 from spinal cord RNA samples obtained from WT ($n = 6$) and KO ($n = 3$) mice. The expression levels were normalized against GAPDH mRNA and expressed as relative fold change with respect to a WT spinal cord sample (mean ± SD) set to a value of 1. Statistical analyses were performed using Student $t$ test. (B) sqRT-PCR showing the expression of lncMN3 transcript on RNA from WT, hemizygous, and homozygous mice spinal cord. The upper band corresponds to the amplification of unedited transcript, while lower band corresponds to the amplification of the transcript bearing the deletion of 253 nucleotides encompassing the ORF. GAPDH was used as a loading control. Representative experiment of three biological replicates. (C) Western blot with anti-SERTM2 antibodies on lysates from WT and hemizygous mouse spinal cord immunoprecipitated with anti-SERTM2 antibody. Input sample accounts for 1.5% of the extract. ACTN was used as a negative control. Representative experiment of three biological replicates. (D) Treadmill test performance represented by shock numbers per run on the following four groups of 3-month-old mice: WT male ($n = 10$), HEM males ($n = 12$), WT female ($n = 9$), HOMO female ($n = 9$). Treadmill test was repeated twice a week for a total of five runs (I–V) and shock numbers were recorded. Mean values ± SD are shown. (E) Number of shocks received in the fifth run by each group of mice described in (D). The box plots show the maximum and minimum values at the whiskers, the 75th and 25th percentiles at the boxes, and the median at the central line. Statistical analyses were performed using one-way ANOVA followed by Bonferroni's multiple comparison test. The adjusted $P$ values are indicated in the right panel. Source data are available online for this figure.

following the evaluation of the coding potential through bioinformatic tools and the detection of the flagged sORF product, validation of the micropeptide's production was achieved, both in mouse and human MNs, using an antibody specific to the micropeptide. Concerning the functional characterization, we observed, in vitro, that SERTM2-KO motor neurons show a slight defect in neuronal excitability due to a depolarized membrane resting potential. As neuronal excitability is only slightly affected, we were not surprised that, in vivo, a phenotype was present only when mice were subjected to an exhaustion test. In particular, in SERTM2-KO motor neurons both spontaneous firing and synaptic transmission rates were not affected, but in the presence of larger depolarizing stimuli the excitability was reduced (as shown in Fig. 4C). This lower excitability in the absence of the SERTM2

micropeptide, likely due to a higher percentage of inactivated voltage-dependent channels at more depolarized resting membrane potentials (Angsutararux et al, 2021), becomes functionally detectable only at high frequency of excitatory signaling, suggesting that the micropeptide contributes to maintain excitability during demanding conditions, coherently with the behavioral observations. Exercise is indeed commonly described as brain and spinal cord plasticity modulator (Gardiner et al, 2006; Ang and Gomez-Pinilla, 2007; Woodrow et al, 2013) and it has been associated with changes in neurotrophic factors production and electrophysiological properties (Chopek et al, 2015). Gardiner et al (2006) and Power et al (2018) described how exercise improve motor neuron excitability, depending on the type and intensity of training and on the type of motor neurons involved. Moreover, among the

parameters affected by training, a hyperpolarization of the resting membrane potential was observed (Beaumont and Gardiner, 2003). Those considerations allow us to speculate that we observed a phenotype only during exercise, because in that moment there is a demand for a higher neuronal excitability that cannot be experienced by SERTM2-KO motor neurons.

We then wanted to understand in more detail the molecular mechanism underlying the observed phenotypes. Initially, we sought to identify characterized proteins showing a resemblance to SERTM2. From the sequence homology analysis, it emerged that SERTM2 bears a striking resemblance to a family of type I transmembrane peptides encoded by the five KCNE genes, well-known for their involvement in regulating potassium channels (Appendix Fig. S2), suggesting that SERTM2 could be involved in the modulation of this type of channels. Specifically, the conservation of amino acids observed between SERTM2 and three KCNE peptides (1, 2, and 3) pertains to the transmembrane domain, which, in the case of the latter, has been hypothesized to be responsible for determining the modulation specificity of KCNQ1 channels (Melman et al, 2001). However, other studies have shown that the carboxy-terminal domain, conserved between KCNE1 and KCNE3, is responsible for the modulation of KCNQ1 (Tapper and George, 2000). Subsequently, Gage and Kobertz (2004) highlighted how the sole transmembrane domain of KCNE3 is necessary for its modulatory activity on KCNQ1, proposing a bipartite model for the modulation of KCNQ1 by KCNE1 and KCNE3 subunits. In the proposed model, it has been suggested that the transmembrane domain of KCNE3 takes an active role in modulation, superseding the contribution of the COOH terminus, while the transmembrane domain of KCNE1 is passive, revealing COOH-terminal modulation of KCNQ1 channels. Gage and Kobertz (2004) also observed that truncating most of the NH2 terminus doesn't significantly change Q1–E3 basal activation as long as a single putative N-linked glycosylation site remains intact. It is interesting to note that SERTM2 also features a putative N-linked glycosylation site in its N-terminal domain (Asn11), further enhancing the analogy between these regulatory systems. As shown in the manuscript, SERTM2 can interact with the two-pore domain potassium channel TASK1, that together with TASK3 is one of the main determinants of resting membrane potential and motor neurons excitability (Enyedi and Czirják, 2010; Berg et al, 2004). Interestingly, the interaction between SERTM2 and TASK1 can be modeled on the interaction between KCNE3 and KCNQ1. Therefore, considering the structural similarity between these pairs, one could hypothesize a functional mechanism involving the SERTM2 transmembrane domain similar to that observed for KCNE3, though further experiments are necessary to precisely unravel this mechanism.

Another hypothesis of SERTM2 mechanism of action come from the intrinsic characteristic of TASK1 channels. These channels serve as background K+ channels: when located on the plasma membrane, they are constitutively active and allow the movement of potassium ions. However, their activity is regulated in multiple ways, including post-translational modification and trafficking (Mathie et al, 2010). Binding of chaperone 14-3-3 masks the ER retention signal on TASK1 and enables its trafficking to the plasma membrane (Zuzarte et al, 2009) while p11 and COP1 are respectively involved in TASK1 retrograde transport and retention to the ER (Renigunta et al, 2006; Zuzarte et al, 2009) suggesting that the number of channels present on the cell surface is tightly regulated.

In this view, the modulation of TASK1 activity by SERTM2 could be exerted through TASK1 retention in the membrane. Indeed, the micropeptide itself could mask interaction sites with other proteins known and capable of mediating trafficking.

All our work conducted in motor neurons has focused on the role of SERTM2 micropeptide, as we demonstrated that the phenotype observed in KO-MNs can be attributed to the absence of the micropeptide rather than to the absence of the transcript, but we cannot exclude the possibility that the transcript might have its own function.

About this concept, it is interesting to note that beyond its expression in the spinal cord, we found good levels of lncMN3 transcript in the 3-month-old mouse lung where the micropeptide is undetectable. Those observations suggest that in this specific tissue the role of the SERTM2 gene could be exerted by the lncMN3 noncoding counterpart.

In addition, a recent paper describes the role of lncMN3, renamed CARDEL (Cardiac Development Long noncoding RNA), in human cardiac specification and in cardiomyocyte physiological function, leaning toward the hypothesis that the micropeptide is not produced (Pereira et al, 2024). However, the analyses performed by Pereira et al on the micropeptide existence in human are only predictive, while the WB on extract derived from iPSCs induced to MN differentiation allowed us to show that also the human sequence can be translated, suggesting that the phenotype described in cardiomyocyte should be further investigated to identify which is the functional product of the SERTM2 gene, that at present remains bifunctional.

In this work, we have demonstrated the existence and the role of SERMT2, a micropeptide derived from a previously annotated noncoding transcript, emphasizing the importance of studying the activity of the coding part of this class of RNAs especially in the context of the central nervous system where this feature is underestimated. Given the impact of ion channels on neuronal physiology, we believe that the example of SERTM2 should be paradigmatic for a class of fine regulators that can modulate neuronal properties by acting on ion channels.

## Methods

### Reagents and tools table

| Reagent/resource | Reference or source | Identifier or catalog number |
|---|---|---|
| Trypsin solution from porcine pancreas | Sigma-Aldrich | cat# T4549 |
| Trypsin-EDTA Solution | Sigma-Aldrich | cat# T4299 |
| EmbryoMax DMEM | Millipore | cat# SLM-220-B |
| Neurobasal™ Medium | Gibco | cat# 21103049 |

| Reagent/resource | Reference or source | Identifier or catalog number |
| --- | --- | --- |
| Advanced DMEM/F12 | Gibco | cat# 12634010 |
| Dulbecco's Modified Eagle's Medium/Nutrient Mixture F12 Ham | ThermoFisher Scientific | cat# D6421 |
| DMEM- High glucose | Sigma-Aldrich | cat# D6546 |
| Sterile Earle's Balanced Salt Solution (EBSS) | Sigma-Aldrich | cat# E7510 |
| Embryonic stem-cell FBS, qualified, US origin | Gibco | cat# 16141-079 |
| KnockOut™ Serum Replacement | ThermoFisher Scientific | cat# 10828028 |
| Horse serum | ThermoFisher Scientific | cat# 16050122 |
| Fetal Bovine Serum, qualified, Brazil | Gibco | cat# 10270-106 |
| Penicillin/Streptomycin | Sigma-Aldrich | cat# P0781 |
| GlutaMAX™Supplement | ThermoFisher Scientific | cat# 35050061 |
| L-glutamine | Sigma-Aldrich | cat# G7513 |
| EmbryoMax non-essential a.a. | Millipore | cat# TMS-001-C |
| EmbryoMax Nucleosides (100X) | Millipore | cat# ES-008-D |
| 2-mercaptoethanol for ES cells | Millipore | cat# ES-007-E |
| D-(+)-Glucose solution | Sigma-Aldrich | cat# G8769 |
| B-27™Supplement (50X), serum free | ThermoFisher Scientific | cat# 17504001 |
| N-2 supplement | Gibco | cat# 17502-001 |
| ESGRO® Recombinant Mouse LIF Protein | Chemicon | cat# ESG11107 |
| GSK-3 Inhibitor XVI | Sigma-Aldrich | cat# 361559 |
| PD173074 | Sigma-Aldrich | cat# P2499 |
| Smoothened agonist, SAG | Sigma-Aldrich | cat# 566660 |
| Retinoic Acid | Sigma-Aldrich | cat# R2625 |
| L-ascorbic acid | Sigma-Aldrich | cat# TMS-001-C |
| Recombinant Human GDNF | Peprotech | cat# 450-44 |
| Recombinant Human CNTF | Peprotech | cat# AF-450-13 |
| Recombinant Human/Murine/Rat BDNF | Peprotech | cat# 450-02 |
| Y-27632 dihydrochloride | Sigma-Aldrich | cat# Y0503 |
| Deoxyribonuclease I from bovine pancreas | Sigma-Aldrich | cat# DN25 |
| Poly-L-ornithine | Sigma-Aldrich | cat# P-3655 |
| Murine Laminin | Sigma-Aldrich | cat# L2020 |
| Ultrapure Water with 0.1% Gelatin | Millipore | cat# ES-006-B |
| Opti-MEM™ Reduced Serum Medium | ThermoFisher Scientific | cat# 31985070 |
| Lipofectamine™ 2000 Transfection Reagent | ThermoFisher Scientific | cat# 11668019 |
| WesternBright ECL kit | Advansta | cat# K-12045-D20 |
| cOmplete™, EDTA-free Protease Inhibitor Cocktail | Roche | cat# 11873580001 |
| NuPAGE 4–12% Bis-Tris Gel | Invitrogen | cat# NP0321 |
| Mini-PROTEAN TGX | Bio-Rad | cat# 4568083 |
| Amersham Protran 0.45 µm Nitrocellulose Blotting Membrane | Cytiva | cat# 10600002 |
| Difco Skim Milk | BD Life Sciences | cat# 232100 |
| 4X Laemmli Sample Buffer | Bio-Rad | cat# 161-0747 |
| Bio-Rad Protein Assay Dye Reagent Concentrate | Bio-Rad | cat# 5000006 |
| Dynabeads Protein G | ThermoFisher Scientific | cat# 10004D |
| DAPI for nucleic acid staining | Sigma-Aldrich | cat# D9542 |
| Paraformaldehyde | Electron Microscopy Sciences, Hatfield, PA | cat# 15710 |
| Donkey serum | Sigma-Aldrich | cat# D9663 |

| Reagent/resource | Reference or source | Identifier or catalog number |
| --- | --- | --- |
| Goat serum | Sigma-Aldrich | cat# G9023 |
| Mem-PER Plus Membrane Protein | ThermoFisher scientific | cat# 89842 |
| Nutristem | Sartorius | cat# 05-100-1A |
| Dispase | Gibco | cat# 17105041 |
| Geltrex | ThermoFisher Scientific | cat# A1413202 |
| ProLong™ Diamond Antifade Mountant | ThermoFisher scientific | cat# P-36961 |
| Image-iT™ FX Signal Enhancer | ThermoFisher scientific | cat# I36933 |
| Puromycin | Invivo Gen | cat# ant-pr-1 |
| HCR Buffers | Molecular Instruments | N/A |
| HCR Amplifier (647 fluorophore) | Molecular Instruments | N/A |
| **Experimental models** | | |
| HBG3 ES cell line carrying an Hb9-GFP transgene | Provided by Prof. Niel A. Shneider (Columbia University) | N/A |
| CF1 Mouse Embryonic Fibroblasts, irradiated | Gibco | cat# A34180 |
| HeLa | ATCC | cat# CCL-2 |
| Mouse C57BL/6J, SERTM2 $-/-$ | This paper | N/A |
| IPSC NIL | Provided by Prof. Alessandro Rosa (Sapienza University) | N/A |
| **Recombinant DNA** | | |
| pSERTM2 | This paper | N/A |
| pSERTM2-FLAG | This paper | N/A |
| pTASK1 | This paper | N/A |
| pTASK1-FLAG | This paper | N/A |
| pDONOR-FLag | This paper | N/A |
| pDONOR-KO | This paper | N/A |
| **Antibodies** | | |
| Anti-SERTM2 | Thermofisher | Custom (Project: 2XB1182-AB4385) |
| Anti-HPRT | Santa Cruz | cat# sc-376938 HRP |
| Anti-actinin | Santa Cruz | cat# sc-390205 |
| Anti-FLAG | Sigma-Aldrich | cat# F1804 |
| Anti-TASK1 | Alomone | cat# APC-024 |
| Anti-CALR | Cell Signaling Technology | cat# 2891 |
| Goat anti-rabbit | Invitrogen | cat# 31460 HRP |
| Goat anti-mouse | Invitrogen | cat# 32430 HRP |
| Goat anti-Mouse IgG Alexa Fluor 488 | Thermofisher | cat# A11001 |
| Donkey anti-Rabbit IgG Alexa Fluor Plus 647 | ThermoFisher Scientific | cat# A32795 |
| Rabbit IgG Isotype Control | Invitrogen | Cat# 02-6102 |
| Anti MAP2 | Abcam | Cat# Ab5392 |
| **Oligonucleotides and other sequence-based reagents** | | |
| lnc MN-3 mouse F: | AGAAACGGGCATCAGAAACTG | |
| lnc MN-3 mouse R | ATGAGTACGGCTGGCGATTT | |
| GAPDH mouse F | TGACGTGCCGCCTGGAGAAA | |
| GAPDH mouse R | AGTGTAGCCCAAGATGCCCTTCAG | |
| HB9 mouse F | TGCCAGCACCTTCCAACT | |
| HB9 mouse R | CTTCCCCAAGAGGTTCGACT | |
| ATP5O mouse F | CAACCGCCCTGTACTCTGCT | |

| Reagent/resource | Reference or source | Identifier or catalog number |
|---|---|---|
| ATP5O mouse R | GGATTCAGAACAGCCAGAGACAC | |
| Oligo Race | GAACGCTTTCATGACGGCAA | |
| pSERTM2-FLAG FW cloning | AGCGTTTAAACTTAAGCTcttgagtg ATGACGGAGGTA | |
| pSERTM2-FLAG RV cloning | CGAGCTCGGTACCAAGCTTTACTT GTCGTCATCGTCTTTGTAGTCAGGA GTGGGAATTCGACT | |
| pSERTM2 FW cloning | TAAAGCTTGGTACCGAGCTCGGAT | |
| pSERTM2 RV cloning | AGGAGTGGGAATTCGACTTTGCAT | |
| pTASK1 FW cloning | aaaaaGGATCCatgaagcggcagaacgtgcg | |
| pTASK1 RV cloning | aaaaaGAATTCagcgcaggagctccgtgtga | |
| pTASK1-FLAG FW cloning | gatgacgacaagGAATTCTGCAGATATCCAGC | |
| pTASK1-FLAG RV cloning | gtctttgtagtcTCACACGGAGCTCCTGCGCT | |
| Inverse pcr FW | CAGGATTATGCTGAGACTTGGCG | |
| Inverse pcr RV | TTACTGTCGTCATCGTCTTTGTAG | |
| FW br2 flag | CGATGACGACAGTAAAGTCAGGATTA TGCTGAGACTTGG | |
| RV br2 flag | CTCAGCATAATCCTGTGTGTTGTGTG TGAAGCAATTC | |
| FW ctrl gen 1 | TCTTGAGTGATGACGGAGGT | |
| RV ctrl gen 1 | GCAGAATGACGAGCATCAAA | |
| FW arm1 | aaaagagctcTTTAAATACCACCGCGCCCTG | |
| RV arm1 | aaaaggtaccAGTCACAGAATGAGTACGGCT | |
| FW arm 2 | aaaaggatccCAACTCCCACTTGTGCAAGC | |
| RV Arm 2 | aaaaggatccCCGACACCAAGAGGGATCAC | |
| FW ctrl gen 2 | TCTTGAGTGATGACGGAGGT | |
| RV ctrl gen 2 | GCAGAATGACGAGCATCAAA | |
| **Chemicals, enzymes, and other reagents** | | |
| FastDigest BamHI | ThermoFisher Scientific | cat# FD0054 |
| FastDigest EcoRI | ThermoFisher Scientific | cat# FD0274 |
| 10X FastDigest Green Buffer | ThermoFisher Scientific | cat# B72 |
| PrimeScript-RT Master Mix | Takara Bio | cat# RR036b |
| PowerUp SYBR-Green Master Mix | ThermoFisher Scientific | cat# 4385612 |
| Mytaq DNA polymerase | Bioline | cat# BIO-21105 |
| CloneAmp™ HiFi PCR Premix | Clontech | cat# 639298 |
| RNase inhibitors | Thermo Fischer Scientific | cat# EO0384 |
| Proteinase K, recombinant PCR Grade | Roche | cat# 03115828001 |
| Papain | Worthington Biochemical Corporation | cat# LK003176 |
| Ovomucoid inhibitor-Albumin | Worthington Biochemical Corporation | cat# LK003182 |
| Triton X-100 | Sigma-Aldrich | cat# 9002-93-1 |
| Dulbecco's Phosphate Buffered Saline w/o MgCl2 and CaCl2 | Sigma-Aldrich | cat# D8537 |
| Dulbecco's Phosphate Buffered Saline with MgCl2 and CaCl2 | Sigma-Aldrich | cat# D8662 |
| Direct-Zol RNA MiniPrep Kit | Zymo Research | cat# R2050 |
| NuceloSpin Gel and PCR Clean-up | MACHEREY-NAGEL | cat# 740609.250 |
| Plasmid DNA extraction Mini kit | Fisher Molecular Biology | cat# DE-035 |
| NucleoBond Xtra Midi EF | MACHEREY-NAGEL | cat# 740420.50 |
| Genomic DNA Extraction Kit mini | RBC Real Genomics | cat# YGB50 |

| Reagent/resource | Reference or source | Identifier or catalog number |
|---|---|---|
| FirstChoice® RLM-RACE Kit | Ambion | cat# AM1700 |
| In-Fusion® HD Cloning Kit | Takara Bio | cat# 102518 |
| Cycloheximide | Sigma | cat# C7698-5G |
| miRNeasy Mini Kit | Qiagen | cat# 217004 |
| PCRBIO Rapid exctract pcr kit | PCRBIOSYSTEMS | cat# PB10.24-40 |
| **Software** | | |
| ImageLab | Bio-Rad | https://www.bio-rad.com/it-it/product/image-lab-software?ID=KRE6P5E8Z |
| Prism 9 | GraphPad by Dotmatics | https://www.graphpad.com/scientific-software/prism/ |
| QuantStudio 3 and 5 RealTime PCR System Software | ThermoFisher Scientific | https://www.thermofisher.com/it/en/home/global/forms/life-science/quantstudio-3-5-software.html |
| MetaMorph Microscopy Automation and Image Analysis Software | Molecular Devices | RRID:SCR_002368 https://www.andImageAnalysisSoftware moleculardevices.com/products/cellular-imaging-systems/acquisition-and-analysis-software/metamorphmicroscopy#gref |
| ImageJ, Fiji distribution | ImageJ | https://imagej.net/software/fiji/downloads |
| NIS-Elements AR software | Nikon | https://www.microscope.healthcare.nikon.com/en_EU/products/software/nis-elements/nis-elements-advanced-research |
| CPC | Kong et al, 2007 | http://cpc2.cbi.pku.edu.cn/ |
| CPAT | Wang et al, 2013 | http://lilab.research.bcm.edu/cpat/index.php |
| PhyloCSF | Lin et al, 2011 | http://www.broadinstitute.org/compbio1/PhyloCSFtracks/trackHub/hub.txt |
| PRALINE | Bawono and Heringa J, 2014 | https://www.ibi.vu.nl/programs/pralinewww/ |
| Protter | Omasits et al, 2014 | http://wlab.ethz.ch/protter |
| deepTMHMM | Hallgren, et al, 2022 bioRxiv | https://dtu.biolib.com/DeepTMHMM |
| I-TASSER | Yang et al, 2015 | http://zhanglab.ccmb.med.umich.edu/I-TASSER |
| Alphafold2 webservices | Jumper et al, 2021 | https://www.alphafoldserver.com |
| AlphaFold3 | Abramson et al, 2024 | https://www.alphafoldserver.com |
| chopchop | Labun et al, 2019 | https://chopchop.cbu.uib.no/ |
| pClamp 10 software | Molecular Devices, Union City, CA | https://support.moleculardevices.com/s/article/Axon-pCLAMP-10-Electrophysiology-Data-Acquisition-Analysis-Software-Download-Page |
| TIS Predictor | Gleason et al, 2022 | https://www.tispredictor.com |
| GWIPS-viz Browser | Kiniry et al, 2018 | https://gwips.ucc.ie/cgi-bin/hgGateway |
| **Other** | | |
| Subcloning Efficiency DH5α Competent Cells | Invitrogen | cat# 18265-017 |
| MAX Efficiency DH5α Competent Cells | Invitrogen | cat# 18258-012 |

## Cell cultures conditions and treatments

All cell lines used in this study were grown in an incubator at 37 °C, 5% $CO_2$, and routinely tested for mycoplasma contamination.

Murine HBG3 ES cells (embryonic stem cells derived from HB9::GFP transgenic mice- mESCs HB9::GFP) were cultured as described in Pellegrini et al, 2023. Briefly, mESCs HB9::GFP were cultured on gelatin-coated or MEF-coated treated dishes and maintained in mESC medium (Dulbecco's Modified Eagle's

Medium for ES, 15% Fetal Bovine Serum for ES, 1× GlutaMAX, 1× Non-essential Ammino Acids, 1× nucleosides, 1×2-mercaptoethanol and 1× Penicillin/Streptomycin) supplemented with LIF (103 unit/mL), FGFRi (1.5 µM) and Gsk-3i (1.5 µM) (LIF+2i condition). Medium was changed daily and cells were passaged every 2–3 days with 1× Trypsin-EDTA solution. Spinal motor neurons (MNs) were differentiated from mESCs HB9::GFP according to Wichterle et al (2002) and Errichelli et al (2017). At the stage of EBs 6, papain dissociation was performed and cells were plated on polyornithine/laminin-coated dishes with N2B27 medium (50% DMDM F12, 50% Neurobasal, 1× Glutamax, 1× Penicillin–Streptomycin, 1× B-27, 1× N-2, 200 ng/mL Ascorbic Acid, 20 ng/mL BDNF, 10 ng/mL CNTF, 10 ng/mL GDNF, 10 nM Rhok inhibitor) and maintained in culture for the indicated days.

HeLa cells were cultured in Dulbecco's Modified Eagle's Medium (10% fetal bovine serum, 1× Glutamine and 1× Penicillin/Streptomycin). Transient transfection of overexpression plasmids was carried out by using Lipofectamine 2000 (Thermo-Fisher) according to the manufacturer's instructions. The complete medium was replaced after 5 h from transfection.

The human NIL-iPSC line, generated by inserting a cassette for the ectopic expression of transcription factors Ngn2, Isl1, and Lhx3 (NIL) into the genome via a piggyBac transposon vector, as described in Garone et al (2019), was maintained in Nutristem (Sartorius, #05-100-1 A) supplemented with 0.1% Pen/Strep on geltrex-coated plates (ThermoFisher Scientific, #A1413202). The cells were passaged every 4–5 days using 1 mg/ml Dispase (Gibco). Motor neuron differentiation was induced by the ectopic expression of the aforementioned transcription factors, as described in Garone et al (2019), and MN were maintained in culture for the indicated number of days.

## RNA extraction and quantification by sqRT-PCR and qRT-PCR

Total RNA was isolated using Directzol RNA MiniPrep (Zymo Research), with a 15 min on-column DNase treatment. Extracted RNA was retrotranscribed with PrimeScript-RT Reagent Kit (Takara Bio) according to the manufacturers' instructions. sqRT-PCR was performed using MyTaq™ HS Red Mix (Bioline). qRT-PCR was performed using PowerUp SYBR-Green Master Mix (A25742, Life Technologies). DNA amplification was monitored with an ABI 7500 Fast qPCR instrument. Quantitative data were presented as the relative expression of the interested genes normalized to that of GAPDH using the $2 − \Delta\Delta Ct$ comparative method. Oligonucleotides used for sqRT-PCR and qRT-PCR are listed below:

> lnc MN-3 mouse F: AGAAACGGGCATCAGAAACTG
> lnc MN-3 mouse R: ATGAGTACGGCTGGCGATTT
> GAPDH mouse F: TGACGTGCCGCCTGGAGAAA
> GAPDH mouse R: AGTGTAGCCCAAGATGCCCTTCAG
> HB9 mouse F: TGCCAGCACCTTCCAACT
> HB9 mouse R: CTTCCCCAAGAGGTTCGACT
> ATP5O mouse F: CAACCGCCCTGTACTCTGCT
> ATP5O mouse R: GGATTCAGAACAGCCAGAGACAC.

## RNA and protein extraction from mouse tissues

Mice were sacrificed by cervical dislocation. Total RNA from liquid nitrogen-powdered tissues homogenized in Qiazol reagent (QIAGEN) was obtained using Directzol RNA MiniPrep kit (Zymo Research) according to the manufacturer's instructions.

For protein extraction, tissue samples were washed in PBS, weighed, cut into small pieces and homogenized in a dounce with a teflon pestle in ice-cold RIPA buffer containing protease inhibitors (Complete, EDTA-free, Roche). After 15 strokes the homogenate was incubated on a rotator for 30 min at 4 °C and then spun at $12,000 \times g$ for 10 min at 4 °C. The supernatant was collected and quantified with the Bradford colorimetric reaction (Biorad).

## 3′ RACE

RACE PCR was performed using "FirstChoice RLM-RACE Kit, Ambion, AM1700", following manufacturer's instructions on RNA obtained from Embryoid bodies at day 6 of differentiation. Oligonucleotide F used in combination with reverse oligo present in the kit has the following sequence: GAACGCTTTCATGACGGCAA.

## Polysome profiling

Polysome profiling was performed as follow: EB cultures at day 6 were washed twice with ice cold 1× PBS containing 100 µg/mL of cycloheximide and resuspended in Polysome lysis buffer (10 mM Tris pH 7.5, 100 mM NaCl, 10 mM $MgCl_2$, 0.5% Triton X-100, and 0.5% sodium deoxycholate) supplemented with protease inhibitors (Complete, EDTA-free, Roche) and 1× RNase inhibitor (Invitrogen). Cell extracts were incubated 10 min on ice and centrifugated 5 min at $12,000 \times g$ at 4 °C. Supernatants from lysed cells were carefully loaded onto 15–50% sucrose gradients and centrifuged at 37,000 rpm with a SW41 rotor (Beckman) for 90 min at 4 °C. Fractions were collected with a Bio-logic LP (Biorad) collector. RNA was extracted using RNeasy Mini Kit (Qiagen) according to the manufacturer's instructions.

## Protein extraction

Whole-cell protein extracts were obtained using RIPA buffer. Membrane-cytosol protein separation was performed with the Mem-PER Plus Membrane Protein Extraction Kit (ThermoFisher Scientific), according to the manufacturer's instructions.

## Western blot

Protein extracts were loaded on NuPAGE SDS-PAGE (Invitrogen) or Mini-PROTEAN TGX (Bio-Rad) precast acrylamide gels according to the manufacturer's instructions. They were then transferred to Amersham Protran 0.45-mm nitrocellulose membrane (GE Healthcare Life Sciences) in Transfer Buffer (25 mM TRIS, 192 mM glycine, 20% methanol). Membranes blocked with 5% nonfat dry milk (Difco skim milk) for 1 h were then incubated overnight at 4 °C with the following primary antibodies: anti-SERTM2 (Custom, Thermofisher, raised against 72:89 residues); anti-HPRT (sc-376938 HRP); anti-actinin (sc-390205); anti-FLAG (Sigma-Aldrich F1804); anti-TASK (Alomone, APC-024); anti-CALR (Cell Signaling Technology #2891). The following secondary antibodies were used: goat anti-rabbit HRP (31460, Invitrogen) and goat anti-mouse HRP (32430, Invitrogen). Protein detection was carried out with WesternBright ECL (Advansta) using Chemi-DocTM MP System.

## Co-IP

Cells were washed twice with ice-cold PBS (Sigma-Aldrich), gently scraped with lysis buffer (50 mM Tris-HCl pH 7.5, 150 mM NaCl, 20 mM NaF, 10 mM NaVO4, 1%NP-40, 1%Tryton100, 1% SDS) or RIPA buffer supplemented with 1× protease inhibitor (Complete, EDTA-free, Roche). Lysates were incubated on a rotator for 60 min at 4 °C, spun down at 13'000× rpm for 10 min at 4 °C, and then supernatants were collected. 1 mg of extract was diluted in 1 ml of co-IP buffer (50 mM Tris-HCl pH 7.5, 150 mM NaCl, 1 mM EDTA, 0.25% NP-40, 5% glycerol) supplemented with 1× protease inhibitor (Complete, EDTA-free, Roche), precleared (2 h at 4 °C on a rotator) and incubated either with 5γ of primary antibody or control IgG overnight at 4 °C on a rotator. 10% or 1.5% of the extract was saved to be used as input. The next day, 40 µl of pre-washed Dynabeads protein G (ThermoFisher Scientific) were added to each sample. Samples were then incubated on a rotator for 4 h at 4 °C. The beads were recovered through a magnetic rack and washed four times for 5 min on a rotator at room temperature with ice-cold 500 µl of co-IP buffer. After the last wash, beads were resuspended in Laemmli buffer 1× (Biorad) supplemented with 1 mM DTT and heated for 10 min at 95 °C. Variable fractions of input and IP or IgG samples were loaded for western blot analysis.

## Immunofluorescence and FISH

Cells were fixed for 20 min at RT with cold 4% paraformaldehyde (Electron Microscopy Sciences, #15710) diluted in complete PBS and stored in PBS at 4 °C. Cells were permeabilized and blocked with 0.1% Triton X-100 (Sigma-Aldrich, #X100), goat serum 10% (#G9023 Sigma-Aldrich), glycine 0.3%, diluted in complete PBS for 30 min at room temperature. Subsequently, cells were incubated with Image-iT™ FX Signal Enhancer (ThermoFisher Scientific, #I36933) for 30 min at room temperature, and after washes with PBS, primary antibodies anti-FLAG (Sigma-Aldrich F1804); anti-SERTM2 (Custom, Thermofisher); anti-TASK (Alomone, APC-024); diluted in blocking solution (10% goat serum/Triton 100 × 0.1% in PBS) were added overnight at 4 °C. Secondary antibodies (Goat anti-Mouse IgG Alexa Fluor 488, #A11001; Donkey anti-Rabbit IgG Alexa Fluor Plus 647 ThermoFisher Scientific #A32795) diluted in goat serum 1% were incubated for 45 min at room temperature. Nuclei were stained with 1 mg/ml DAPI (#D9542, Sigma-Aldrich) diluted in complete PBS for 5 min and coverslips were mounted with ProLong™ Diamond Antifade Mountant (Thermo Fischer Scientific, # P-36961).

RNA-FISH staining was carried out with the HCR RNA-FISH technology in accordance to manufacturer protocol (https://www.molecularinstruments.com/hcr-rnafish), with specific modifications to improve simultaneous RNA and protein visualization.

Confocal images were acquired with an inverted Olympus iX73 microscope equipped with a Crest Optics X-Light V3 spinning disk head and BSI Prime sCMOS camera (Photometrics). All images were acquired as 16 bit depth by using a UPLANSApo 60X oil objective (NA 1.35) and collected with MetaMorph software (Molecular Devices). Stacks of images were taken automatically with 0.2 micron between the Z-slices. All Z stacks were merged with maximum intensity projection and combined in a multicolor image. Digital enlargements were obtained by manual cropping from original full-field images. Post-acquisition processing was performed by FIJI tools and in particular: "unsharp mask" was used to enhance the signal over background, while "color balance" was manually adjusted to set image background. Colocalization between FLAG-SERTM2/TASK1 signals was quantified as Pearson's correlation index (PCC) on ROIs on maximum Z-projection by JaCoP plug-in tool. Linescan analyses were performed by plot-profile in FIJI software.

## Plasmid overexpression

### SERTM2 and TASK1 flag cloning

The construct for the overexpression of SERTM2-FLAG (pSERTM2-FLAG) was obtained using the In-Fusion HD cloning kit (Takara Bio USA, Inc.). SERTM2 sequence was amplified by PCR from WT mice spinal cord cDNA using the oligonucleotides: 5'-AGCGTTTAAACT-TAAGCTcttgagtgATGACGGAGGTA-3' and 5'-CGAGCTCGGTAC-CAAGCTTTACTTGTCGTCATCGTCTTTGTAGTCAGGAGTGG-GAATTCGACT -3' and inserted in pCDNA3.1(+) plasmid (Invitrogen). Subsequently a second plasmid, without the Flag sequence downstream SERTM2 (pSERTM2) was obtained through inverse PCR with the oligonucleotides: 5'-TAAAGCTTGGTACCGAGCTCGGAT-3' and 5'-AGGAGTGGGAATTCGACTTTGCAT-3'.

The construct for the overexpression of TASK1-Flag was obtained in two steps by cloning first the TASK1 sequence in pCDNA3.1(+) plasmid (Invitrogen) (pTASK1). The TASK1 sequence was amplified by PCR from WT mice spinal cord cDNA using the oligonucleotides: 5'-aaaaaGGATCCatgaagcggcagaacgtgcg-3' and 5'-aaaaaGAATTCagcgcag-gagctccgtgtga-3'. The uppercase bases are BamHI and EcoRI restriction sites. The flag was inserted by reverse PCR using the following oligonucleotides: 5'-gatgacgacaagGAATTCTGCAGATATCCAGC-3' and 5'-gtctttgtagtcTCACACGGAGCTCCTGCGCT-3' to generate a plasmid containing the flagged sequence of Task1 (pTASK1-FLAG).

## mESCs genome editing

Flag knock-in clones were generated using CRISPR/CAS9 genome-editing system. Single guide RNAs (sgRNAs) targeting downstream SERTM2-ORF sequence were designed using the CRISPR design tool (https://chopchop.cbu.uib.no/ Labun et al, 2019), and the favorite one was cloned into the pX330 vector (Addgene#42230)

>sgRNA: AGGGGTCCTTACACATTCAA

To allow the homology-directed repair, a donor plasmid (pDONOR-FLAG) was obtained from the pSERTM2-FLAG (described above). The ORF sequence of the pSERTM2-FLAG corresponded to the left homology arm, while the right homology arm was cloned through the amplification of the genomic sequence after SERTM2 stop codon, through the In-Fusion HD cloning kit (Takara Bio USA, Inc.).

Starting plasmid was opened with an inverse PCR using the following oligonucleotides:

Inverse PCR FW: CAGGATTATGCTGAGACTTGGCG

Inverse PCR RV: TTACTGTCGTCATCGTCTTTGTAG.

The genomic sequence was amplified using the following oligonucleotides:

FW br2 flag: CGATGACGACAGTAAAGTCAGGATTATGCT GAGACTTGG

RV br2 flag: CTCAGCATAATCCTGTGTGTTGTGTGTGAA GCAATTC.

mESCs were transfected on gelatin-coated dishes using reverse transfection with lipofectamine2000 (Life Technologies) following the manufacturer's instruction and transfecting a total of 1.5 µg of DNA (PX330+sgRNA and pDONOR) for $2.5 \times 10^5$ cells.

After 2 days, mESC were tripsinized and diluted in a 96-well in order to obtain single clones. Clones were then passaged in 48 and 24 wells and gDNA was extracted using PCRBIO Rapid Extract PCR Kit (PCRBIO). Genomic amplification using the oligos listed below revealed clones in which the insertion of the FLAG sequence took place.

CTRL gene 1 FW: TCTTGAGTGATGACGGAGGT
CTRL gene 1 RV: GCAGAATGACGAGCATCAAA.

LncMN3 knockout clones were generated using CRISPR/CAS9 genome-editing system. Single guide RNAs (sgRNAs) targeting lncMN3 exon2 sequence were designed using the CRISPR design tool (https://chopchop.cbu.uib.no/; Labun, et al, 2019) and cloned into the pX330 vector (Addgene#42230)

>sgRNA1: AGTCACAGAATGAGTACGGC.

The homology-directed repair construct (pDONOR-KO) was obtained from HR110 Pa-1plasmid (System Bioscience) that contains a floxed puromycin resistance gene flanked by two multiple cloning sites (MCS1 and MCS2) in which left and right homology arms were cloned. A Poly(A)/2×MAZ sequence (as described in Pellegrini et al, 2023) was cloned downstream the left homology arm.

Left and right homology arms were obtained by PCR from mESC gDNA using the following oligonucleotides:

>FW arm1: aaaagagctcTTTAAATACCACCGCGCCCTG
>RV arm1: aaaaggtaccAGTCACAGAATGAGTACGGCT
>FW arm 2: aaaaggatccCAACTCCCACTTGTGCAAGC
>RV Arm 2: aaaaggatccCCGACACCAAGAGGGATCAC.

mESCs were transfected on gelatin-coated dishes using reverse transfection with lipofectamine2000 (Life Technologies) following the manufacturer's instruction and transfecting a total of 1.5 µg of DNA (PX330+sgRNA and pDONOR) in $2.5 \times 10^5$ cells.

Selection in 0.5 µg/ml puromycin allowed to isolate the colonies that had integrated the selection cassette. After two days mESCs were tripsinized and diluted in a 96-well to obtain single clones. Clones were then passaged in 48 and 24 wells and gDNA was extracted using PCRBIO Rapid Extract PCR Kit (PCRBIO). Genomic amplification revealed clones in which the insertion of the PAS took place.

To obtain ΔORF SERTM2 clones, a guide RNA targeting SERTM2-ORF sequence (chrX:142,945,670-142,945,944, GRCm39/mm39 assembly) was designed using the same tool described above.

>sgRNA: TCATGGAAATCTCACTGGAC.

mESCs were transfected with 1 µg of sgRNA plasmid, using the Lipofectamine 2000 (ThermoFisher). After two days mESCs were tripsinized and diluted in a 96-well to obtain single clones. Clones were then passaged in 48 and 24 wells and gDNA was extracted using PCRBIO Rapid Extract PCR Kit (PCRBIO). Genomic amplification was performed using the oligos listed below:

CTRL gene 2 FW: TCTTGAGTGATGACGGAGGT
CTRL gene 2 RV: GCAGAATGACGAGCATCAAA.

Sanger sequencing revealed clones in which frameshift mutation took place.

## Patch clamp

Electrophysiological experiments were performed on GFP-positive motor neurons derived from HB9::GFP-mESCs using the patch-clamp technique in the whole-cell configuration. Recordings were obtained using a HEKA EPC800 amplifier, Digidata 1322 A analog-to-digital converter, and pClamp 10 software (Molecular Devices, Union City, CA). Data were filtered at 2 kHz and digitized at 5 kHz. During recordings, cells were continuously perfused using a gravity-driven perfusion system with the following external solution: 140 mM NaCl, 2.8 mM KCl, 2 mM $MgCl_2$, 2 mM $CaCl_2$, 10 mM HEPES, and 10 mM glucose (pH 7.4; 300 mOsm). The internal pipette solution contained: 140 mM KCl, 5 mM BAPTA, 2 mM Mg-ATP, and 10 mM HEPES (pH 7.4; 300 mOsm). Borosilicate glass pipettes were pulled with a Narishige puller to a typical pipette resistance of 3–4 MΩ. Evoked action potentials were recorded from mESC-derived neurons applying increasing depolarizing current steps (1 s, 2–20 pA per step) in current-clamp configuration. Spontaneous action potentials were recorded in current-clamp configuration without current injection. Spontaneous synaptic activity was recorded in voltage-clamp configuration at a holding potential of −70 mV.

## Molecular modeling

The interaction between SERTM2 (Uniprot ID: SRTM1_HUMAN) and TASK1 (Uniprot ID: KCNK3_HUMAN) was predicted using the AlphaFold3 server. A pTM = 0.6 was obtained (this measures the accuracy of the entire structure. A pTM score above 0.5 means the overall predicted fold for the complex might be similar to the true structure (Zhang and Skolnick, 2004; Xu and Zhang, 2010). Five obtained complexes were compared with the three-dimensional structure of TASK1 (PDB: 6RV3), revealing a minimum RMSD of 0.8 Å. AlphaFold3 predictions were then cross-referenced with the DALI server to identify structurally similar complexes in the PDB.

## Mouse model

Mice were housed in the animal facility of EMBL Monterotondo at the Gene Editing and Embryology Facility (GEEF). Mice were maintained in temperature and humidity-controlled condition with food and water provided ad libitum and on 12-h light–dark cycle (light on at 7:00). They were housed in IVC Thoren racks in group of 4 mice/cage. All experiments were approved by the Italian Ministry of health (approval n.82945.56) and conducted within the animal welfare regulations and guidelines.

SERTM2 knockout mice were generated in the C57BL/6J background using a CRISPR genome editing system. Two guides (crRNA#1: CAAGTCTTGAGTGATGACGG and crRNA#3: TAATCCTGACTTTAAGG), targeting the region of interest upstream and downstream the SERTM2-sORF, were designed using online tool (https://chopchop.cbu.uib.no/; Labun et al, 2019) and delivered in fertilized eggs together with an in vitro transcribed Cas9 mRNA and a ssODN donor oligonucleotide (ssODN). ssODN was composed of 75 nt of homology arms upstream the SERTM2 ATG codon, followed by a sequence corresponding to the first 11 nucleotides of SERTM2 that were mutated to insert a stop codon in frame with a disrupted ATG codon. Downstream the described 11nt the ssODN continue with 75 nt of homology arms corresponding to the downstream STOP codon sequence (see Fig. EV5B). Editing was verified by PCR genotyping. gDNA extraction from tail biopsies was performed using RBC Real

Genomics DNA Extraction Kit (RBC Bioscience) according to the manufacturer's protocol.

Behavioral and functional tests were conducted during the light phase of the light/dark cycle and. Before testing, animals were habituated to the testing room for at least 30 min. The treadmill exhaustion tests were performed as described in Pellegrini et al (2023).

## Single-cell analysis

Single-cell RNA expression matrix was retrieved from GSE174671. Data were collected and processed using Seurat package (version 5.0.0) as described in Carvelli et al (2022). Count matrix related to cells belonging to "WT" condition was normalized using "NormalizeData" function with standard parameters (log transform of gene counts normalized by cell-library size). mRNA encoding for channels were identified using MGI description retrieved using Biomart (https://www.bioconductor.org/packages/release/bioc/html/biomaRt.html). Then co-expression metric was calculated using Pearson's correlation comparing lncMN3 (A730046J19Rik) expression to RNA levels of each channel transcripts.

## In silico tools

LncMN3 coding potential was assessed using CPC (https://cpc2.gao-lab.org/), CPAT (http://lilab.research.bcm.edu/cpat/) and PhyloCSF (Mudge et al, 2019) online tools. The alignment of mouse and human predicted SERTM2 micropeptides was performed using PRALINE algorithm (https://www.ibi.vu.nl/programs/pralinewww/; Bawono and Heringa J, 2014. Protter (Omasits et al, 2014), deepTMHMM (Hallgren, et al, 2022 bioRxiv), I-TASSER (Yang et al, 2015) and Alphafold2 webservices (Jumper et al, 2021) were used to predict SERTM2 topology.

Translation initiation site prediction and Kozak similarity scores were retrieved using the online tool TIS predictor (https://www.tispredictor.com/).

Ribosome profiling (Ribo-seq) and associated RNA-seq data were explored in the GWIPS-viz browser (http://gwips.ucc.ie/), an online genome browser for viewing ribosome profiling data. *Global aggregate* track represent data from all available human and mouse studies.

## Quantification and statistical analysis

The data are presented as mean ± SEM or mean ± SD, as specified in the figure legends, along with the number of biological replicates, P values and descriptions of the graph used. The statistical tests used to evaluate differences between means are detailed in the figure legends. Patch-clamp data were analyzed using one-way ANOVA or the Kruskal–Wallis nonparametric test, depending on the data distribution. In cases where statistical significance was detected ($P < 0.05$), post hoc multiple comparison procedures were performed using the Holm–Sidak method for parametric data or Dunn's method for nonparametric data. Significance for all tests was set at $P < 0.05$. All statistical analyses were performed using GraphPad Prism (version10.4.1).

## Graphics

The synopsis image was created with BioRender.com.

## Data availability

This study includes no data deposited in external repositories.

The source data of this paper are collected in the following database record: biostudies:S-SCDT-10_1038-S44319-025-00404-w.

## Peer review information

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

## Acknowledgements

This work is dedicated to Dr. Silvia Biscarini, whose passion for science and teaching will continue to inspire us all. The authors would like to thank N Humphreys and M Ascolani from the Gene Editing and Embryology Facility (GEEF) at EMBL- Rome for the help in the generation of the mutant mouse line; Dr M Caruso for assistance; Prof. Alessandro Rosa for providing iPS-NIL cells; PhD Valeria de Turris at the Center for Life Nano & Neuro Science Imaging Facility, Istituto Italiano di Tecnologia, for technical advice on image acquisition. This work was partially supported by grants from: ERC-2019-SyG 855923-ASTRA, NextGeneration EU PNRR MUR -"National Center for Gene Therapy and Drugbased on RNA Technology" (CN00000041) to IB,PRIN MUR 2022 n. 2022HM5LFW to JM, Consiglio Nazionale delle Ricerche - CNR (project DBA.AD005.225-NUTRAGE-FOE2021) to JM, NextGeneration EU  PNRR MUR M6C2 - Investment 2.1 Enhancement and strengthening of biomedical research in the NHS (PNRR-MAD-2022-692 12376434) to SF and TD.

## Author contributions

**Michela Lisi**: Conceptualization; Investigation; Visualization; Writing—original draft. **Tiziana Santini**: Investigation. **Tiziano D'andrea**: Investigation. **Beatrice Salvatori**: Investigation. **Adriano Setti**: Formal analysis. **Alessandro Paiardini**: Formal analysis. **Sofia Nutarelli**: Investigation. **Carmine Nicoletti**: Investigation. **Flaminia Pellegrini**: Investigation. **Sergio Fucile**: Supervision; Funding acquisition. **Irene Bozzoni**: Supervision; Funding acquisition; Writing—original draft; Writing—review and editing. **Julie Martone**: Conceptualization; Supervision; Funding acquisition; Investigation; Visualization; Writing—original draft; Writing—review and editing.

Source data underlying figure panels in this paper may have individual authorship assigned. Where available, figure panel/source data authorship is listed in the following database record: biostudies:S-SCDT-10_1038-S44319-025-00404-w.

## Disclosure and competing interests statement

The authors declare no competing interests.

# Expanded View Figures

**Figure EV1.  Characterization of lncMN3 expression profile.**                                                          ▶

(**A**) UMAP plot of the integrated dataset from Carvelli et al (2021) depicting cell identity assignment to NP, MNP, EMN, LMN, IN, or NA (cells that could not be assigned to any specific identity) subpopulations. (**B**) Single-cell expression of lncMN3 and marker genes over the UMAP representation of the integrated dataset. (**C**) qRT-PCR showing the expression levels of lncMN3 in spinal cord RNA samples obtained from mice at the indicated ages. The expression levels were normalized against the GAPDH mRNA and expressed as relative fold change with respect to a P3 sample set to a value of 1. The mean ± SD of 3 mice for each group is shown. (**D**) Left panel: UMAP visualization of single-cell RNA sequencing data from mouse spinal cord. Cell types of interest are color-coded according to their identity: orange for progenitor of motor neurons (pMN) and violet for motor neurons (MN). Middle panel: Expression levels of lncMN3 (A730046J19Rik) in the UMAP plot derived from mouse spinal cord single-cell RNA sequencing. lncMN3 expression levels are represented with violet-yellow color scale: higher intensity (yellow) corresponds to higher expression levels, while lower intensity (violet) indicates lower expression levels. Right panel: Dotplot depicting lncMN3 (A730046J19Rik) expression levels in mouse spinal cord cell populations. Y-axis depicts identified cell populations while x-axis depicts for each cell population the fraction of cells expressing the gene. LncMN3 expression levels are represented with yellow-red color scale: higher intensity (dark red) corresponds to higher expression levels, while lower intensity (yellow) indicates lower expression levels. The expression of lncMN3 in MN is highlighted by the orange box. (**E**) Left panel: UMAP visualization of single-cell RNA sequencing data from human spinal cord. Cell types of interest are color-coded according to their identity: lilac for progenitor of motor neurons (pMN) and green for motor neurons (MN). Middle panel: Expression levels of lncMN3 (SERTM2) in the UMAP plot derived from human spinal cord single-cell RNA sequencing. LncMN3 expression levels are represented with violet-yellow color scale: higher intensity (yellow) corresponds to higher expression levels, while lower intensity (violet) indicates lower expression levels. Right panel: Dotplot depicting lncMN3 (SERTM2) expression levels in human spinal cord cell populations. Y-axis depicts identified cell populations while x-axis depicts for each cell population the fraction of cells expressing the gene. LncMN3 expression levels are represented with yellow-red color scale: higher intensity (dark red) corresponds to higher expression levels, while lower intensity (yellow) indicates lower expression levels. The expression of lncMN3 in MNs is highlighted by the orange box.

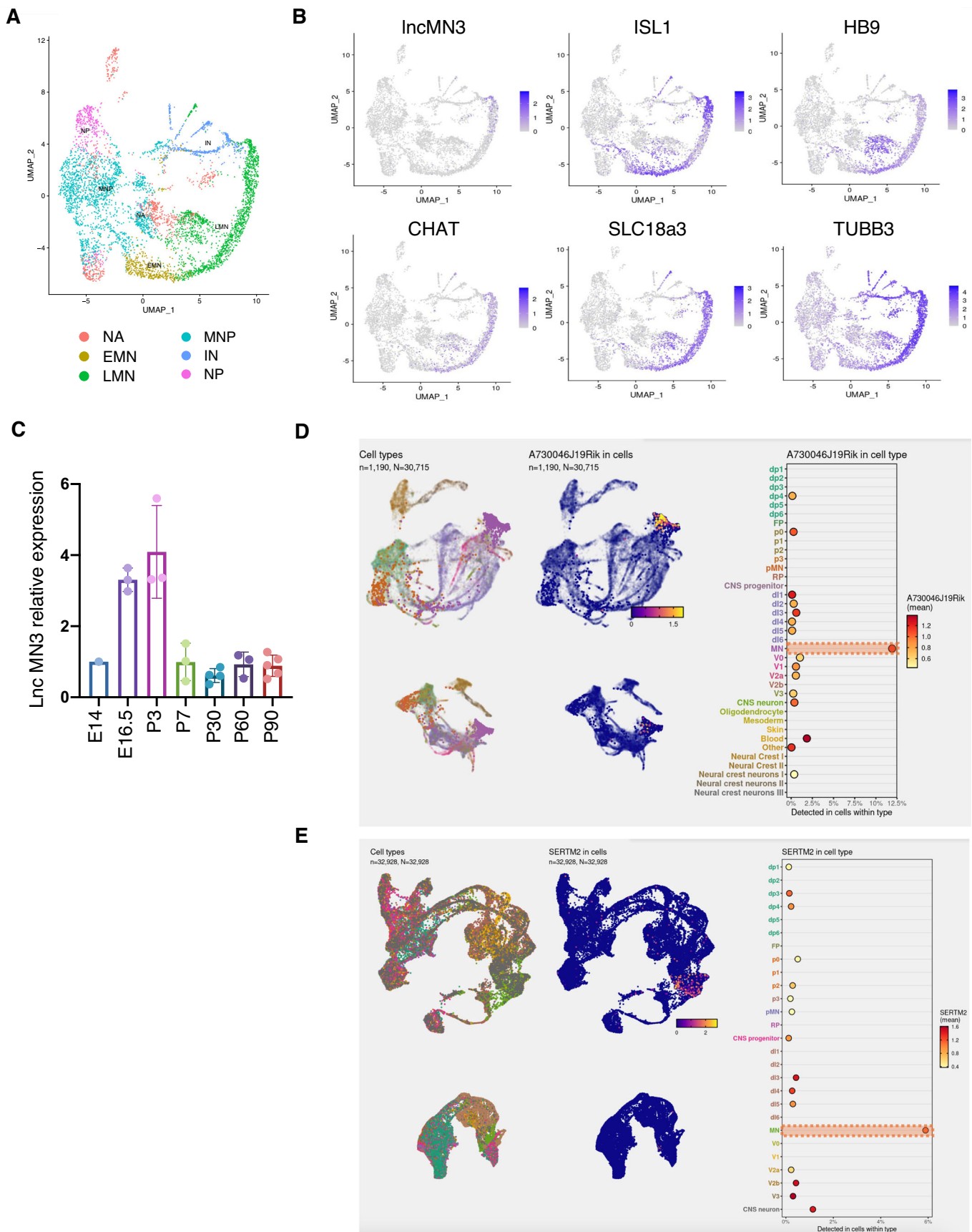

**A**

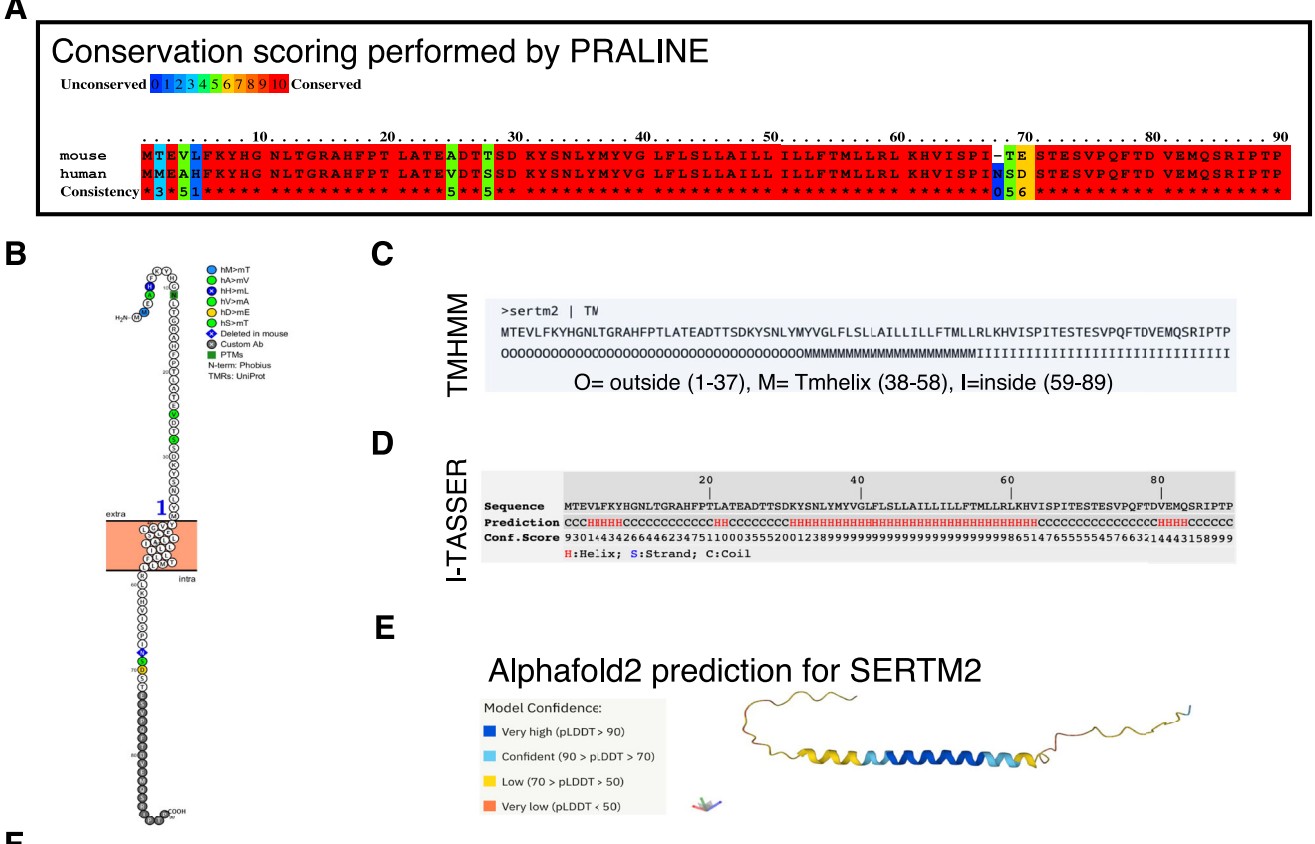

Conservation scoring performed by PRALINE

Unconserved 0 1 2 3 4 5 6 7 8 9 10 Conserved

**B**

**C**

TMHMM

>sertm2 | TM

MTEVLFKYHGNLTGRAHFPTLATEADTTSDKYSNLYMYVGLFLSLLAILLILLFTMLLRLKHVISPITESTESVPQFTDVEMQSRIPTP
OOOOOOOOOOOOOOOOOOOOOOOOOOOOOOOOOOOOOOMMMMMMMMMMMMMMMMMMMMMIIIIIIIIIIIIIIIIIIIIIIIIIIIIIIIII

O= outside (1-37), M= Tmhelix (38-58), I=inside (59-89)

**D**

I-TASSER

**E**

Alphafold2 prediction for SERTM2

Model Confidence:
- Very high (pLDDT > 90)
- Confident (90 > p.DDT > 70)
- Low (70 > pLDDT > 50)
- Very low (pLDDT < 50)

**F**

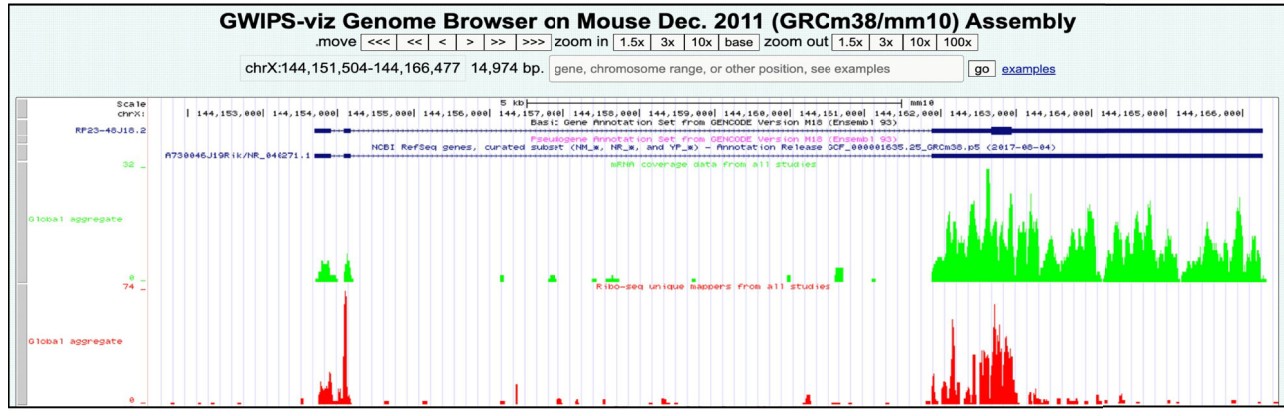

**G**

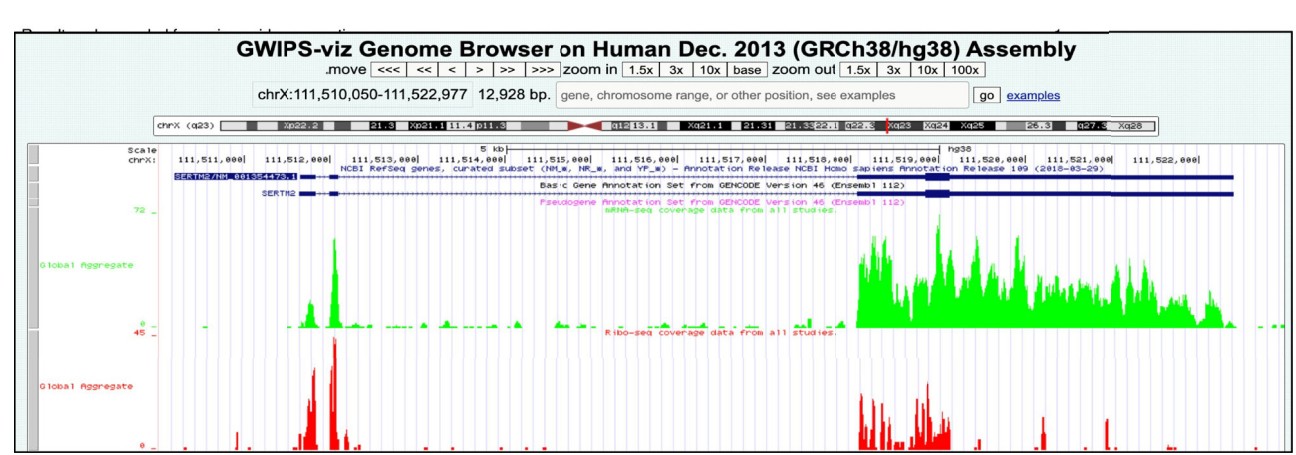

◀ **Figure EV2.  LncMN3 Coding potential.**

(**A**) Alignment of mouse and human predicted lncMN3-derived micropeptides performed using PRALINE algorithm (Praline scores range: 0–10). (**B**–**E**) A conserved transmembrane (TM) domain is predicted in both species by different algorithms (PROTTER—**B**; TMHMM—**C**; TASSER—**D**; and Alphafold2—**E**). (**B**) The SERTM2 sequence recognized by the custom antibody is depicted in gray. Amino acid substitutions between human and mouse sequences are also indicated with the same color code used in (**A**). (**F**) Screenshot of ribosome profiling analysis using the GWIPS-viz browser for mouse A730046j19Rik (lncMN3) locus. Red histogram represents ribo-seq coverage data from all studies (Elongating Ribosomes-Footprints). Green histogram represents mRNA-seq coverage data from all studies (mRNA-seq Reads). (**G**) Screenshot of ribosome profiling analysis using the GWIPS-viz browser for human SERTM2 locus. Red histogram represents ribo-seq coverage data from all studies (Elongating Ribosomes-Footprints). Green histogram represents mRNA-seq coverage data from all studies (mRNA-seq Reads).

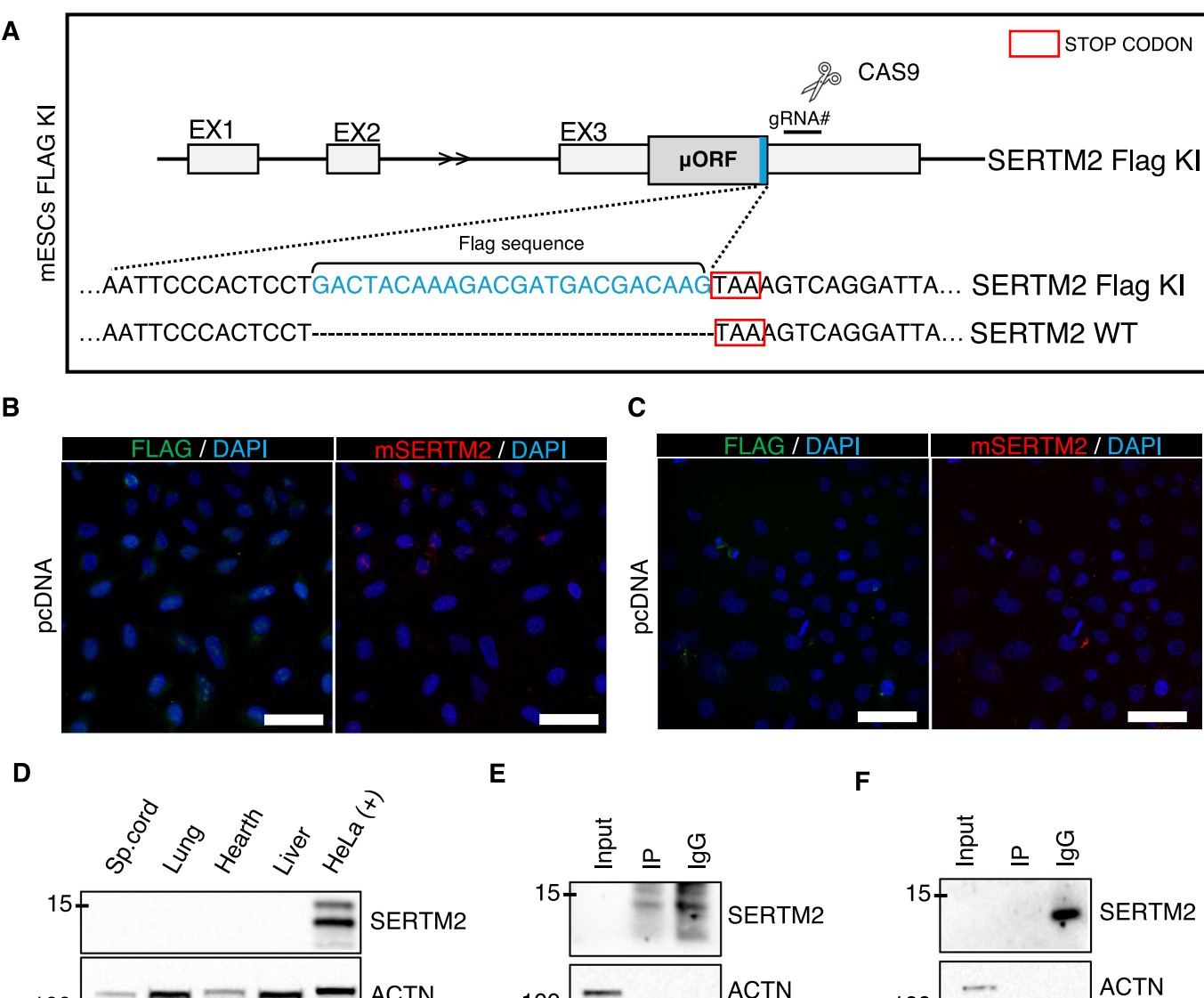

**Figure EV3.  SERTM2 is a type I transmembrane micropeptide.**

(**A**) Schematic representation of CRISPR/CAS9 genome editing strategy used to insert a C-terminal FLAG-tag in frame with the sORF in mESCs. (**B**) Immunofluorescence for FLAG (green signal, left panel) and SERTM2 (red signal, right panel) detection performed on HeLa cells transfected with an empty plasmid (pcDNA). Before immunofluorescence, samples were permeabilized with (0.1%) Triton X-100 detergent. Nuclei were stained with DAPI (blue signal). Scale bar corresponding to 50 µm. Representative experiment of 3 biological replicates. (**C**) Immunofluorescence for FLAG (green signal, left panel) and SERTM2 (red signal, right panel) detection performed on HeLa cells transfected with an empty plasmid (pcDNA). Triton X-100 permeabilization treatment was omitted. Scale bar corresponding to 50 µm. Representative experiment of 3 biological replicates. (**D**) Western blot analyses using antibodies against SERTM2 performed on protein extracts obtained from the indicated tissues. A protein extract obtained from HeLa overexpressing SERTM2-FLAG was used as positive control. ACTN was used as loading control. Representative experiment of 3 biological replicates. (**E**) Representative western blot with anti-SERTM2 antibodies on proteins obtained after SERTM2 immunoprecipitation from 3-month-old mouse liver extract (P90). Input sample accounts for 1.5% of the extract. ACTN was used as negative control. Representative experiment of 2 biological replicates. (**F**) Representative western blot with anti-SERTM2 antibodies on proteins obtained after SERTM2 immunoprecipitation from 3 months old mouse lung extract (P90). Input sample accounts for 1.5% of the extract. ACTN was used as negative control. Representative experiment of 3 biological replicates.

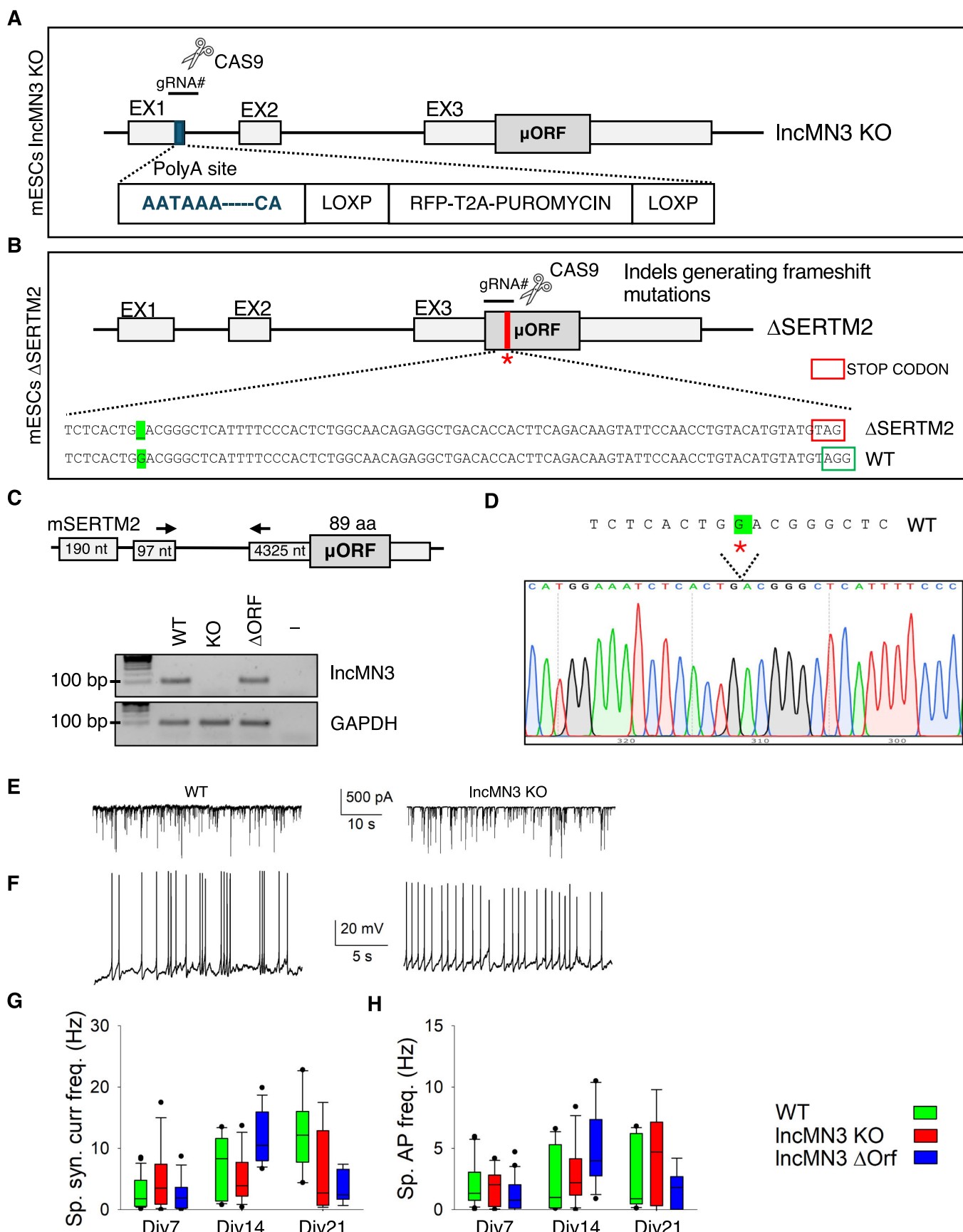

**A** mESCs lncMN3 KO

CAS9
gRNA#

EX1 EX2 EX3 µORF lncMN3 KO

PolyA site

AATAAA-----CA | LOXP | RFP-T2A-PUROMYCIN | LOXP

**B** mESCs ΔSERTM2

gRNA# CAS9 Indels generating frameshift mutations

EX1 EX2 EX3 µORF ΔSERTM2

STOP CODON

TCTCACTG**G**ACGGGCTCATTTTCCCACTCTGGCAACAGAGGCTGACACCACTTCAGACAAGTATTCCAACCTGTACATGTAT**GTAG** ΔSERTM2
TCTCACTG**G**ACGGGCTCATTTTCCCACTCTGGCAACAGAGGCTGACACCACTTCAGACAAGTATTCCAACCTGTACATGTAT**GTAGG** WT

**C**

mSERTM2

190 nt | 97 nt | 4325 nt | 89 aa µORF

100 bp — lncMN3
100 bp — GAPDH

WT KO ΔORF –

**D**

T C T C A C T G **G** A C G G G C T C WT

C A T G G A A A T C T C A C T G A C G G G C T C A T T T T C C C

**E** WT lncMN3 KO

500 pA
10 s

**F**

20 mV
5 s

**G**

**H**

WT
lncMN3 KO
lncMN3 ΔOrf

◀ **Figure EV4. Block of SERTM2 expression does not alter the spontaneous activity of mESC-derived neurons.**

(A) Schematic representation of CRISPR/CAS9 genome editing strategy used to obtain lncMN3-KO mESC. (B) Schematic representation of CRISPR/CAS9 genome editing strategy used to obtain ΔSERTM2 mESCs. (C) Upper panel: schematic representation of A730046J19Rik locus. Lower panel: sqRT-PCR showing the expression of lncMN3 transcript on RNA from WT, KO and ΔORF clones. Used oligonucleotide are represented by arrows in the upper panel. GAPDH was used as control. Representative experiment of 3 biological replicates. (D) Sequencing chromatogram of the edited sORF region showing the indel generating the frameshift mutation. WT sequence is indicated above the chromatogram. (E) Typical traces of spontaneous synaptic currents recorded in voltage-clamp configuration at -70 mV from a WT (left) and a lncMN3-KO (right) neuron. (F) Typical traces of spontaneous activity of membrane potential recorded in current-clamp configuration from a WT (left) and a lncMN3-KO (right) neuron. (G) Box and whisker plots representing the frequency of spontaneous synaptic currents of WT, lncMN3-KO and lncMN3-ΔORF neurons, at DIV 7 ($n = 24$, $n = 13$, $n = 20$ for WT, lncMN3-KO and lncMN3-ΔORF, respectively), 14 ($n = 15$, $n = 18$, $n = 13$ for WT, lncMN3-KO and lncMN3-ΔORF, respectively) and 21 ($n = 11$, $n = 8$, $n = 7$ for WT, lncMN3-KO and lncMN3-ΔORF, respectively), as indicated. The box plots display the 90th and 10th percentiles at the whiskers, the 75th and 25th percentiles at the boxes, and the median at the central line. Black circles represent outlier data outside the 10th and 90th percentiles. Statistical analyses were performed using one-way ANOVA. Same cells as Fig. 4. Please note no significant difference. (H) Box and whisker plots representing the frequency of spontaneous APs of WT, lncMN3-KO and lncMN3-ΔORF neurons, at DIV 7 ($n = 24$, $n = 13$, $n = 20$ for WT, lncMN3-KO and lncMN3-ΔORF, respectively), 14 ($n = 15$, $n = 18$, $n = 13$ for WT, lncMN3-KO and lncMN3-ΔORF, respectively) and 21 ($n = 11$, $n = 8$, $n = 7$ for WT, lncMN3-KO and lncMN3-ΔORF, respectively), as indicated. The box plots display the 90th and 10th percentiles at the whiskers, the 75th and 25th percentiles at the boxes, and the median at the central line. Black circles represent outlier data outside the 10th and 90th percentiles. Statistical analyses were performed using one-way ANOVA. Same cells as Fig. 4. Please note no significant difference.

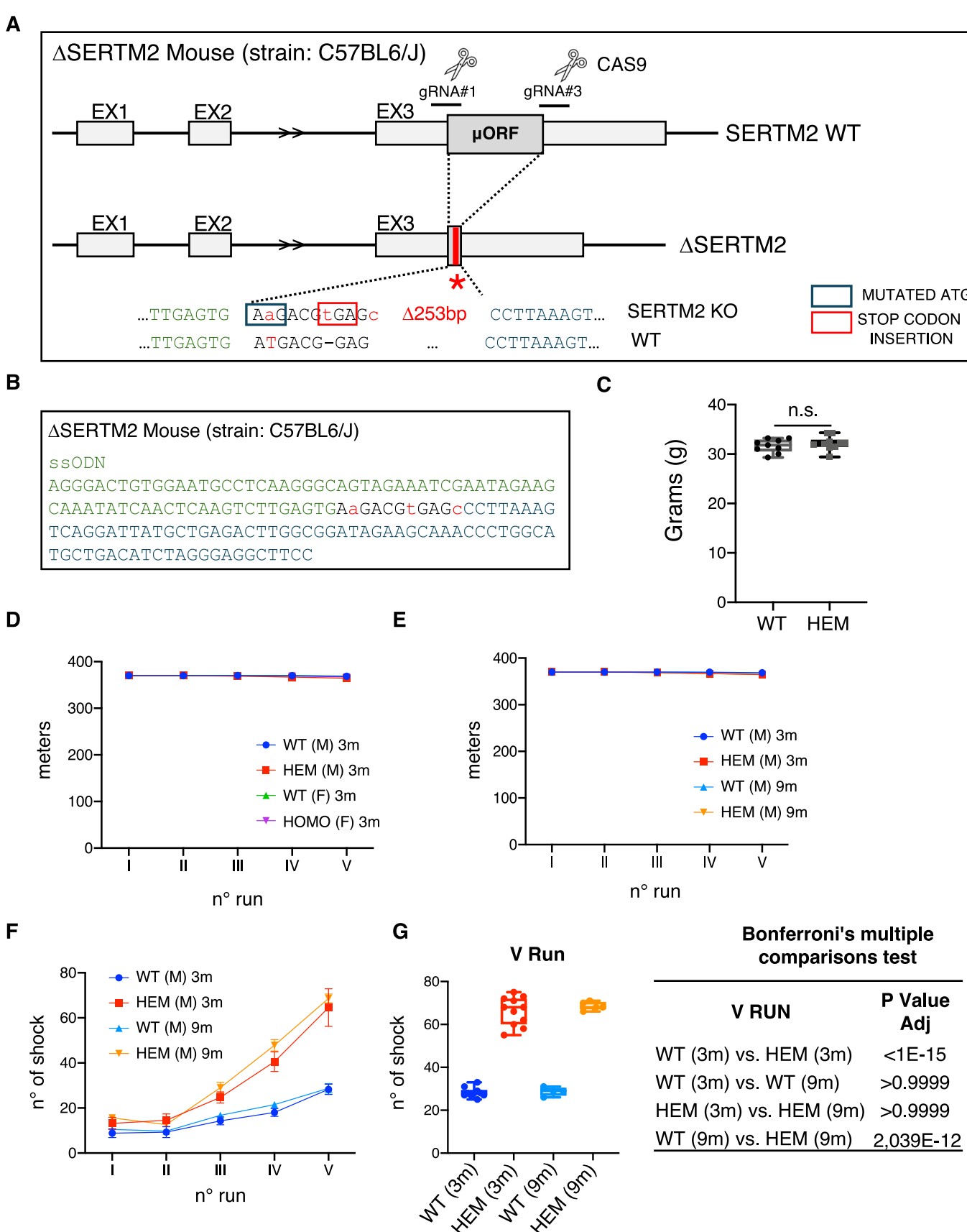

◀ **Figure EV5. SERTM2-KO mouse model characterization.**

(A) Schematic representation of CRISPR/CAS9 genome editing strategy used to obtain ΔSERTM2 mice. (B) ssODN donor sequence used to obtain ΔSERTM2 mice in the CRISPR/CAS9 genome editing strategy. (C) Mean body weight for 3-month-old male WT ($n = 9$) and HEM ($n = 10$) mice. Data are represented as a scatter plot. Each point corresponds to a single value. Statistical analyses were performed using Student $t$ test. (D) Treadmill test performance represented by meters per run on the following 4 groups of 3 months old mice: WT male ($n = 10$), HEM males ($n = 12$), WT female ($n = 9$), HOMO female ($n = 9$). Treadmill test was repeated twice a week for a total of five runs (I–V) and run meters were recorded. Values are mean ± SD. (E) Treadmill test performance represented by meters per run on the following 4 groups of male mice: WT 3 m ($n = 10$), HEM 3 m ($n = 12$), WT 9 m ($n = 4$), HEM 9 m ($n = 5$). Treadmill test was repeated twice a week for a total of five runs (I–V) and run meters were recorded. Values are mean ± SD. (F) Treadmill test performance represented by shock numbers assessed on the following 4 groups of male mice: WT 3 m ($n = 10$), HEM 3 m ($n = 12$), WT 9 m ($n = 4$), HEM 9 m ($n = 5$). Treadmill test was repeated twice a week for a total of five runs (I–V) and shock numbers were recorded. Values are mean ± SD. (G) Number of shocks received in the fifth run by each group of mice described in (F): WT 3 m ($n = 10$), HEM 3 m ($n = 12$), WT 9 m ($n = 4$), HEM 9 m ($n = 5$). The box plots show the minimum and maximum values at the whiskers, the 75th and 25th percentiles at the boxes, and the median at the central line. Statistical analyses were performed using one-way ANOVA followed by Bonferroni's Multiple Comparison test. The adjusted $P$ values are indicated in the right panel.

