## [Peer Review File · EMBO Reports]

SERTM2: a neuroactive player in the world of micropeptides

Michela Lisi, Tiziana Santini, Tiziano D'andrea, Beatrice Salvatori, Adriano Setti, alessandro paiardini, Sofia Nutarelli, Carmine Nicoletti, Flaminia Pellegrini, Sergio Fucile, Irene Bozzoni, and Julie Martone

Corresponding author(s): Julie Martone (julie.martone@cnr.it) , Irene Bozzoni (irene.bozzoni@uniroma1.it)

Review Timeline:

Submission Date:	11th Aug 24
Editorial Decision:	24th Oct 24
Revision Received:	8th Jan 25
Editorial Decision:	27th Jan 25
Revision Received:	11th Feb 25
Accepted:	12th Feb 25

Editor: Esther Schnapp

Transaction Report:

Dear Julie,

Thank you for your patience while your manuscript was peer-reviewed at EMBO reports. We have finally received the full set of referee reports that is pasted below.

As you will see, all referees acknowledge that the findings are interesting. In fact, all of them rate novelty, interest and technical quality of the ms "High", which is rare and great !

As expected, the referees also have several suggestions for how the study could be strengthened, and I think all suggestions are good and should be addressed. Please let me know in case you disagree, and we can discuss the exact revision requirements further, also in a video chat, if you like.

I would thus like to invite you to revise your manuscript with the understanding that the referee concerns must be fully addressed and their suggestions taken on board. Please address all referee concerns in a complete point-by-point response. Acceptance of the manuscript will depend on a positive outcome of a second round of review. It is EMBO reports policy to allow a single round of major revision only and acceptance or rejection of the manuscript will therefore depend on the completeness of your responses included in the next, final version of the manuscript.

We realize that it is difficult to revise to a specific deadline. In the interest of protecting the conceptual advance provided by the work, we recommend a revision within 3 months (24th Jan 2025). Please discuss the revision progress ahead of this time with the editor if you require more time to complete the revisions.

- 1) A data availability section providing access to data deposited in public databases is missing. If you have not deposited any data, please add a sentence to the data availability section that explains that.
- 2) Your manuscript contains statistics and error bars based on $n=2$. Please use scatter blots in these cases. No statistics should be calculated if $n=2$.

3) We replaced Supplementary Information with Expanded View (EV) Figures and Tables that are collapsible/expandable online. A maximum of 5 EV Figures can be typeset. EV Figures should be cited as 'Figure EV1, Figure EV2' etc... in the text and their respective legends should be included in the main text after the legends of regular figures.

5) a complete author checklist, which you can download from our author guidelines <https://www.embopress.org/page/journal/14693178/authorguide>. Please insert information in the checklist that is also reflected in the manuscript. The completed author checklist will also be part of the RPF.

6) Please note that all corresponding authors are required to supply an ORCID ID for their name upon submission of a revised manuscript (<https://orcid.org/>). Please find instructions on how to link your ORCID ID to your account in our manuscript tracking system in our Author guidelines

<<https://www.embopress.org/page/journal/14693178/authorguide#authorshippingguidelines>>

12) All Materials and Methods need to be described in the main text using our 'Structured Methods' format, which is required for all research articles. According to this format, the Methods section includes a separate file Reagents and Tools Table (listing key reagents, experimental models, software and relevant equipment and including their sources and relevant identifiers) and a Methods and Protocols section describing the methods using a step-by-step protocol format. The aim is to facilitate adoption of the methodologies across labs. More information on how to adhere to this format as well as a downloadable template (.docx) for the Reagents and Tools Table can be found in our author guidelines:

An example of a Method paper with Structured Methods can be found here: <https://www.embopress.org/doi/full/10.1038/s44320-024-00037-6#sec-4>

As part of the EMBO publication's Transparent Editorial Process, EMBO reports publishes online a Review Process File (RPF) to accompany accepted manuscripts. This File will be published in conjunction with your paper and will include the referee

reports, your point-by-point response and all pertinent correspondence relating to the manuscript.

I look forward to seeing a revised form of your manuscript when it is ready.

Best wishes,
Esther

Referee #1:

In the manuscript "SERTM2: a neuroactive player in the world of lncRNA-derived micropeptides", Lisi and colleagues identify a micropeptide called SERTM2, which is produced from a long non-coding RNA lncMN3 that is primarily expressed in motor neurons. The researchers confirmed endogenous production of SERTM2 using custom antibodies, demonstrating that the protein is membrane-bound in motor neurons. Notably, the study showed that depletion of SERTM2 protein (rather than lncMN3 RNA) in motor neurons leads to decreased excitability and altered resting membrane potential. Loss of the RNA alone did not produce any notable effects. In vivo, SERTM2 knockout mice exhibited mild motor deficits, and mechanistically, SERTM2 was shown to interact with the TASK1 potassium channel, suggesting its role in regulating motor neuron physiology. Overall, this is solid research with rigorous experimental techniques including FLAG knock-ins, custom antibodies, and lncRNA/protein mES mutants. The study also highlights the broader impact of unannotated micropeptides within the proteome. The manuscript is well-written and will be of interest to the research community. Below are minor experimental and conceptual suggestions to further strengthen the manuscript, focusing primarily on the classification of lncMN3 as a lncRNA versus a protein-coding RNA:

1. After reviewing lncMN3 in human and mouse databases, it appears to be annotated as a protein-coding RNA, not a lncRNA. To avoid potential confusion, I recommend that the authors address lncMN3 classification and clarify whether it should be referred to as a protein-coding RNA or a lncRNA, and whether there are evidences for both.
2. The authors do not address whether lncMN3 has any known non-coding regulatory roles. Furthermore, the effect the authors find seem to be mediated by the protein, not the RNA. Given its classification as a lncRNA, and the conclusion that this is a lncRNA-derived protein, it is important to discuss whether there is any evidence of non-coding functionality or whether its categorization is mis-annotation. Based on evidence presented here, it does look like a misannotated protein coding RNA.
3. Given the findings, I suggest a more precise title. The current title, "SERTM2: a neuroactive player in the world of lncRNA-derived micropeptides", implies dual functionality as both a lncRNA and a micropeptide. However, the data indicate that this might be a protein-coding RNA with no clear RNA-based function.
4. Human Coding Potential Analysis in Figure S2: I recommend moving the human coding potential analysis currently in Figure S2 to the main figures. The PhyloCSF score in this figure clearly indicates a high coding potential. Furthermore, do ribosome profiling data support the association of lncMN3 to ribosomes and its active translation? A quick review of publicly accessible data suggests this is the case, and thus, it would be beneficial to include this information in the manuscript.
5. Can the authors identify a Kozak sequence in the lncMN3 transcript or any other regulatory elements? This would further support the idea that lncMN3 is a functional protein-coding RNA.
6. I suggest smRNA FISH to further confirm the subcellular localization of this RNA. This would help validate whether its distribution is consistent with that of a coding or non-coding transcript (or a transcript with dual functionality).
7. A recent study shows that MALAT1 lncRNA can be selectively trafficked to the cytoplasm of neurons to produce a protein. This is highly relevant in the context of the current work and should be cited in the introduction or discussion. Do the authors hypothesize that lncMN3 has any similar RNA-based role related to subcellular localization, or is it purely a pcRNA (related to my previous comments)? Furthermore, some other lncRNA-derived proteins and lncRNAs with dual functionalities related with subcellular localization, which were found in recent years from well-known lncRNAs, were not discussed (i.e., TUG1).
8. Please list the common synonyms used for lncMN3 in the literature, as this will facilitate cross-referencing across studies.
9. As indicated in Figure S2D, the manuscript identifies a candidate coding region in the lncMN3 transcript that is highly conserved among vertebrates, as evidenced by the PhyloP track hub. I recommend highlighting this region more prominently in the figure.
10. In Figure 3H, the SERTM2 band appears quite intense, suggesting it may be an abundant protein. Do the authors have additional data to quantify SERTM2 levels relative to other proteins in motor neurons?
11. Here, it is not immediately clear to which paper cited above the authors are referring to:
In a previous study (Biscarini et al., 2018), several lncRNAs enriched during in vitro mouse embryonic stem cells (mESCs) differentiation to motoneurons (MNs) were identified (Errichelli et al., 2017; Wichterle et al., 2002). In the paper cited above,

Referee #2:

Lisi et al claim to have identified a new microprotein encoded in a lncRNA expressed in mouse mESCs. They use ectopic expression to characterize cellular localization of the microprotein. They also use CRISPR to deplete the mp in mESCs. In addition, they raised antibodies against the conserved region of the mp and show it is expressed in mouse spinal cord.

functionally, they show the mp affects the activity of motor neurons derived from mESCs. This is through the interaction with the K channel, TASK1/2. They went ahead and generated mp KO mice and found they have reduced excitability.

I find this a very interesting and manuscript. The data is clear and logical. The claims are well supported by the data. It is not surprising that ko mice exhibit a mild phenotype as this is typical for lncRNA and, perhaps, microproteins.

I have no major comments except I think the authors should consider referencing Chen et al (10.1126/science.aay0262) as they opened the field if I am not mistaken.

Referee #3:

In this manuscript, Lisi and colleagues demonstrate the coding capability of lncMN3 and the role of its encoded peptide in neuronal excitability, using primarily mouse ESC-derived MN cultures and several biochemical, electrophysiological and genome editing approaches.

This is a very interesting, very straightforward and clearly written paper. As the authors state, the general belief is that lncRNAs are non-coding, however, studies like this one are starting to reveal that this concept (or at least the annotation of some of them - or many? -) must be revisited. I only have a couple of points that, in my view, should improve the message.

Main comments:

- I don't quite understand why the authors target Pereira et al manuscript in their discussion, given that that study covers the potential role of lncMN3 in human cardiomyocytes (different species and different cell type). Actually, by doing so, they have raised an important point on the evolutionary conservation of lncMN3/SERTM2 function. Most of the work presented in this study is on mouse ESC-derived EBs/MNs, and the only human data provided is on HeLa cells and upon overexpression. But is SERTM2 expressed in human spinal cord/iPSC-derived MNs (any other tissue)? Since the mouse-human comparison is mentioned multiple times in the study, to validate the potential relevance of their findings in the human spinal cord (and/or brain), this should be checked, even if no additional biochemistry/mechanistic data in human spinal cord/iPSC-derived MNs is presented.

- The authors show that lncMN3 expression, in mouse, in vitro and in vivo, peaks early during MN development/neonatal stages and drastically decreases thereafter. While I understand that checking for SERTM2 (peptide) is notoriously more difficult, they have proven to be able to do so by IP. It would be interesting to find out whether the peptide is also primarily expressed during late development or whether the levels remain highly expressed in adulthood, which would then justify the interesting data shown in adult mice (Fig. 6). Otherwise, how would the authors explain that a protein that is only expressed during development has an important role in controlling neural/MN excitability in adulthood? This point should at least be discussed.

Minor comments:

- As part of Fig 1, the authors justify that the decrease in lncMN3 expression during ms ESC-derived MN differentiation could be due to a dilution effect since the EBs are only composed by ~40% MNs, which is true. However, they could easily explain this fact, without the need of postulating, by checking lncMN3 expression in FACS-purified HB9:GFP MNs (same time-points during their differentiation and post EB dissociation).

- The authors claim that lncMN3 expression is enriched in MNs, however the scRNAseq data shown in Fig. S1D lncMN3 is expressed in dorsal spinal cord domains (and ventral to a lesser extent) but actually not in the pMN or MN domains. This statement should be rephrased accordingly.

Several sc or snRNAseq studies on human spinal cord have also been recently published (the authors rightly cite one of them, Rayon et al 2021). The authors could also mention if lncMN3 expression was also detected in those studies, and whether it was specially expressed in MNs.

- The WB in Fig. S3F does not indicate the age of the mice. Could it be that the prot isn't detected in the SC (or any other mouse tissue) because of a wrong developmental time checked (postnatal or even adult tissues)? (Also, Fig. 3L should be J).

- The graphs for Fig 4C-I look odd with some bars showing 3 points, others only 2 and some none. Showing the distribution of all recorded neurons would be better (could be in a violin plot).

In addition, as the authors pointed out, these cultures are normally composed by only ~40% MNs. How did the author know that the recordings were made from MNs and not from any other neural type in the culture?, did they record specifically from HB9:GFP+ cells? If not, the claim that SERTM2 is a major determinant of resting membrane potential in MNs would not be supported, and saying "neurons" would be more appropriate.

- How do the authors reconcile the results shown in Fig. S4 (no difference in the frequency of spontaneous synaptic currents and spontaneous action potentials across ms ESC-derived neuron genotypes) with the clear differences shown in Fig. 4 on

evoked APs? I would have appreciated some mention to this in the discussion.

- Adding labels to Fig. 5B would help to understand what each molecule represents. Similarly, all WBs should indicate the mw for the proteins shown.

EMBOR-2024-60181

Dear Dr Schnapp,

Please find below the point-by-point response to the reviewers' criticisms, along with a description of the main changes made to the text.

Referee #1:

In the manuscript "SERTM2: a neuroactive player in the world of lncRNA-derived micropeptides", Lisi and colleagues identify a micropeptide called SERTM2, which is produced from a long non-coding RNA lncMN3 that is primarily expressed in motor neurons. The researchers confirmed endogenous production of SERTM2 using custom antibodies, demonstrating that the protein is membrane-bound in motor neurons. Notably, the study showed that depletion of SERTM2 protein (rather than lncMN3 RNA) in motor neurons leads to decreased excitability and altered resting membrane potential. Loss of the RNA alone did not produce any notable effects. In vivo, SERTM2 knockout mice exhibited mild motor deficits, and mechanistically, SERTM2 was shown to interact with the TASK1 potassium channel, suggesting its role in regulating motor neuron physiology. Overall, this is solid research with rigorous experimental techniques including FLAG knock-ins, custom antibodies, and lncRNA/protein mES mutants. The study also highlights the broader impact of unannotated micropeptides within the proteome. The manuscript is well-written and will be of interest to the research community. Below are minor experimental and conceptual suggestions to further strengthen the manuscript, focusing primarily on the classification of lncMN3 as a lncRNA versus a protein-coding RNA:

1. After reviewing lncMN3 in human and mouse databases, it appears to be annotated as a protein-coding RNA, not a lncRNA. To avoid potential confusion, I recommend that the authors address lncMN3 classification and clarify whether it should be referred to as a protein-coding RNA or a lncRNA, and whether there are evidences for both.

In our work, we describe the ability of lncMN3 to be translated, but we do not exclude the possibility that the transcript may have a function on its own. Furthermore, its non-coding role is described in the paper by Pereira et al., (2024) which has since been accepted and published in the "Cells" journal. Therefore, we believe that further studies are needed to understand whether the function described by Pereira in cardiomyocytes has been wrongly attributed to the lncRNA or whether the product of the gene locus may act in two ways. The current state of knowledge points to a dual function. We have tried to clarify this concept in the manuscript.

2. The authors do not address whether lncMN3 has any known non-coding regulatory roles. Furthermore, the effect the authors find seem to be mediated by the protein, not the RNA. Given its classification as a lncRNA, and the conclusion that this is a lncRNA-derived protein, it is important to discuss whether there is any evidence of non-coding functionality or whether its categorization is mis-annotation. Based on evidence presented here, it does look like a misannotated protein coding RNA.

We understand the reviewer's point of view in having doubts about the classification of IncMN3. When we selected our "gene of interest", we were studying lncRNAs expressed in motor neurons to characterize their function, and we proceeded with the non coding definition. However, the boundary between coding and non-coding is very thin and often simply depends on which molecule was characterized first. It is certainly true that in our case, the observed phenotype, and therefore the function, are associated with the production of the micropeptide, but it is also true that in the paper by Pereira et al. (2024), it is defined as a lncRNA, and the function is attributed to the transcript and not to its putative coding counterpart. Furthermore, we have evidence that the transcript is present in tissues where we have not been able to identify the micropeptide, leading us to hypothesize that the transcript may have an independent function other than the coding one. For example, in the case of adult lung, we observed transcript levels 2.5 times higher than in spinal cord, and using the same IP procedure, we were unable to visualise the micropeptide, which instead was clearly visible in spinal cord and heart protein extracts. We know that this observation doesn't define the transcript function, but it leaves room for the hypothesis that it could have one. Additionally, as stated in the discussion we can make hypothesis, but we do not have data to establish whether the function observed in cardiomyocytes is mediated by SERTM2 micropeptide or, as Pereira et al. claim, by IncMN3 (renamed CARDEL). We believe that even this is a very interesting point, it falls outside the scope of this manuscript. Thus, based on the data at our disposal, although we cannot exclude the possibility of a mis-annotation issue, we favor the dual-function hypothesis. Indeed, there is increasing evidence highlighting the pleiotropic ability of RNA to perform various functions, and given its length of nearly 5 kb, it is easy to hypothesize that the non coding part of the transcript may have functions beyond the regulation of peptide production.

3. Given the findings, I suggest a more precise title. The current title, "SERTM2: a neuroactive player in the world of lncRNA-derived micropeptides", implies dual functionality as both a lncRNA and a micropeptide. However, the data indicate that this might be a protein-coding RNA with no clear RNA-based function.

Following the reviewer's suggestion, we have changed the title of the manuscript as follows: "SERTM2: a neuroactive player in the world of micropeptides".

4. Human Coding Potential Analysis in Figure S2: I recommend moving the human coding potential analysis currently in Figure S2 to the main figures. The PhyloCSF score in this figure clearly indicates a high coding potential. Furthermore, do ribosome profiling data support the association of IncMN3 to ribosomes and its active translation? A quick review of publicly accessible data suggests this is the case, and thus, it would be beneficial to include this information in the manuscript.

In agreement with the reviewer's suggestion, the phyloCSF tracks have been added to the main figures for both mouse and human species (new Fig 1A and 2A respectively).

To further support the association of IncMN3 with ribosomes, we took advantage of GWIPS-viz, a publicly available browser that provides ribo-seq and corresponding RNA-seq data from studies across multiple species (Kiniry et al. Current Protocols in Bioinformatics, 2018). We focused the analyses on the human and mouse transcriptomes. Unfortunately, the GWIPS-viz browser does not provide data for motor neurons in either human or mouse models, so we considered the global coverage derived from the sum of the different studies for both Ribo-seq and RNA-seq. In line with our experimental results, despite the low expression of IncMN3 in the cell types available in GWIPS-viz, it was possible to detect a signal from the aggregate of ribo-seq data suggesting that

IncMN3 can be associated with elongating ribosomes. We have included the results in the new Figure EV2F-G.

5. Can the authors identify a Kozak sequence in the IncMN3 transcript or any other regulatory elements? This would further support the idea that IncMN3 is a functional protein-coding RNA.

Following the reviewer's request, we searched for a Kozak sequence in the IncMN3 transcript using the online tool TISpredictor to score codons by Kozak context using both human and mouse IncMN3 sequences (<https://www.tispredictor.com>). In both cases, the start ATG codon corresponding to SERTM2 ("position 1081" in human ENST00000569275.2 sequence and "position 1140" in mouse ENSMUST00000135687.2 sequence) was identified and obtained a Kozak similarity score among the highest (Table S1 in Appendix). We added this information to the main text. Interestingly, when we used as positive control the pTUNAR bifunctional transcript (also known as Megamind in Drosophyla), the TISpredictor analysis gave to the start ATG codon corresponding to the TUNAR micropeptide, the same score obtained for SERTM2 (0.79). Those results are consistent with the hypothesis of SERTM2 translation.

6. I suggest smRNA FISH to further confirm the subcellular localization of this RNA. This would help validate whether its distribution is consistent with that of a coding or non-coding transcript (or a transcript with dual functionality).

As suggested by the reviewer, we performed the smRNA FISH experiment to further confirm the subcellular localisation of the IncMN3 transcript and added this information to the manuscript (New Fig 1C). However, we do not necessarily correlate a dual functionality with a dual localization. The fact that a lncRNA is translated does not exclude the possibility that it could also function as a ncRNA in other tissues/conditions in the same subcellular compartment.

7. A recent study shows that MALAT1 lncRNA can be selectively trafficked to the cytoplasm of neurons to produce a protein. This is highly relevant in the context of the current work and should be cited in the introduction or discussion. Do the authors hypothesize that IncMN3 has any similar RNA-based role related to subcellular localization, or is it purely a pcRNA (related to my previous comments)? Furthermore, some other lncRNA-derived proteins and lncRNAs with dual functionalities related with subcellular localization, which were found in recent years from well-known lncRNAs, were not discussed (i.e., TUG1).

We thank the reviewer for the suggestion to include the citation describing the micropeptide derived from a cytoplasmic portion of Malat1. This citation has been added to the Introduction. What we know so far is that IncMN3 is mainly found in the cytoplasm and that it can be translated. Our hypothesis for future work is that the transcript may have other roles related to its RNA nature. Indeed, only a small part of the transcript encodes for the micropeptide (6%), leaving the hypothesis that the remaining sequence could have another function. Furthermore, the fact that it is localised in the cytoplasm does not necessarily mean that it is always translated. For this reason, we have not formulated a functional hypothesis related to the subcellular localisation. For example in the lung, as mentioned above, there is a good level of transcript, but we cannot see the peptide. We believe that in this particular condition, IncMN3 could fulfil its cytoplasmic role by a mechanism that is not related to its translation. These are, of course, only speculations that need to be confirmed experimentally, but they certainly leave plenty of hypotheses for future work.

Regarding the Tug1-derived peptide, it is described in the paper by van Heesch et al. (2019), which was already cited in the introduction. We didn't add the following paper by Lewandowski et al. (2020) regarding the role of the TUG1-derived micropeptide in male fertility because, as the authors state, "there is currently limited evidence for its endogenous existence (of the micropeptide)", so we decided not to add this specific example to the text pending further confirmation.

8. Please list the common synonyms used for IncMN3 in the literature, as this will facilitate cross-referencing across studies.

All the synonymous used for SERTM2 were added in the manuscript

Human: LINC00890; SERTM2; CARDEL

Mouse: A730046J19Rik; IncMN3; SERTM2

9. As indicated in Figure S2D, the manuscript identifies a candidate coding region in the IncMN3 transcript that is highly conserved among vertebrates, as evidenced by the PhyloP track hub. I recommend highlighting this region more prominently in the figure.

As per point 4, we have moved this panel in the main figure and added a yellow box highlighting the conservation of the microORF sequences among vertebrates and the PhyloP track.

10. In Figure 3H, the SERTM2 band appears quite intense, suggesting it may be an abundant protein. Do the authors have additional data to quantify SERTM2 levels relative to other proteins in motor neurons?

Unfortunately, we don't have additional data to verify the absolute abundance of SERTM2 protein or to quantify its relative levels to other proteins (independently from antibodies efficiency).

11. Here, it is not immediately clear to which paper cited above the authors are referring to:

In a previous study (Biscarini et al., 2018), several lncRNAs enriched during in vitro mouse embryonic stem cells (mESCs) differentiation to motoneurons (MNs) were identified (Errichelli et al., 2017; Wichterle et al., 2002). In the paper cited above,

We thank the reviewer for the comment, as the text was indeed unclear. To make it more understandable, we have moved the references of the protocol used to differentiate mESCs into motor neurons (Errichelli et al. (2017) and Wichterle et al. (2002)) to the Materials and methods section.

Referee #2:

Lisi et al claim to have identified a new microprotein encoded in a lncRNA expressed in mouse mESCs. They use ectopic expression to characterize cellular localization of the microprotein. They also use CRISPR to deplete the mp in mESCs. In addition, they raised antibodies against the conserved region of the mp and show it is expressed in mouse spinal cord.

functionally, they show the mp affects the activity of motor neurons derived from mESCs. this is through the interaction with the K channel, TASK1/2. The went ahead and generated mp KO mice and found they have reduced excitability.

I find this a very interesting and manuscript. The data is clear and logical. The claims are well supported by the data. It is not surprising that ko mice exhibit a mild phenotype as this is typical for lncRNA and, perhaps, microproteins. I have no major comments except I think the authors should consider referencing Chen et al (10.1126/science.aay0262) as they opened the field if I am not mistaken.

We thank the reviewer for his suggestion: we have included the reference to the Chen et al. work in the Introduction.

Referee #3:

In this manuscript, Lisi and colleagues demonstrate the coding capability of lncMN3 and the role of its encoded peptide in neuronal excitability, using primarily mouse ESC-derived MN cultures and several biochemical, electrophysiological and genome editing approaches.

This is a very interesting, very straightforward and clearly written paper. As the authors state, the general belief is that lncRNAs are non-coding, however, studies like this one are starting to reveal that this concept (or at least the annotation of some of them - or many? -) must be revisited. I only have a couple of points that, in my view, should improve the message.

Main comments:

- I don't quite understand why the authors target Pereira et al manuscript in their discussion, given that that study covers the potential role of lncMN3 in human cardiomyocytes (different species and different cell type). Actually, by doing so, they have raised an important point on the evolutionary conservation of lncMN3/SERTM2 function. Most of the work presented in this study is on mouse ESC-derived EBs/MNs, and the only human data provided is on HeLa cells and upon overexpression. But is SERTM2 expressed in human spinal cord/iPSC-derived MNs (any other tissue)? Since the mouse-human comparison is mentioned multiple times in the study, to validate the potential relevance of their findings in the human spinal cord (and/or brain), this should be checked, even if no additional biochemistry/mechanistic data in human spinal cord/iPSC-derived MNs is presented.

We share the reviewer's curiosity about lncMN3 translation in humans. We therefore verified the existence of the micropeptide also in human MNs derived from iPSCs and added this information to the paper. These new results have been added to the manuscript (New Figure 3I).

Moreover, taking advantage of this new data and as suggested by the reviewer, we have revised the commentary in the discussion concerning the work of Pereira et al., removing the confusion caused by simultaneously considering different species and tissues, and adding a note about the fact that the human sequence can be translated: 'The Western blot on extracts derived from iPSCs induced to motor neuron differentiation allowed us to demonstrate that the human sequence can also be translated, suggesting that the phenotype described in cardiomyocytes should be further investigated to identify the functional product of the SERTM2 gene....'

- The authors show that lncMN3 expression, in mouse, in vitro and in vivo, peaks

early during MN development/neonatal stages and drastically decreases thereafter. While I understand that checking for SERTM2 (peptide) is notoriously more difficult, they have proven to be able to do so by IP. It would be interesting to find out whether the peptide is also primarily expressed during late development or whether the levels remain highly expressed in adulthood, which would then justify the interesting data shown in adult mice (Fig. 6). Otherwise, how would the authors explain that a protein that is only expressed during development has an important role in controlling neural/MN excitability in adulthood? This point should at least be discussed.

We apologise for the lack of clarity, we have specified in the text that treadmill experiments were performed on three months old mice (P90) and that also the IP shown in former Figure 3I was performed on spinal cord extracts from three months old mice. Thus, the micropeptide is not only produced during embryonic development (former Figure 3L, now Figure 3K) but is also present in the spinal cord of adult mice (former Figure 3I, now Figure 3J). In Figure 1D we show that the expression level of IncMN3 decreases approximately 4-fold between the embryo (E16.5) and three-month-old mice (P90), but still remains expressed at good levels. We have specified the age of the mice more precisely in the two figures (Figures 3 and 6).

Minor comments:

- **As part of Fig 1, the authors justify that the decrease in IncMN3 expression during ms ESC-derived MN differentiation could be due to a dilution effect since the EBs are only composed by ~40% MNs", which is true. However, they could easily explain this fact, without the need of postulating, by checking IncMN3 expression in FACS-purified HB9:GFP MNs (same time-points during their differentiation and post EB dissociation).**

We understand the point raised by the reviewer. We have tried in the past to assess whether IncMN3 is still enriched in the GFP+ population at later times of differentiation. However, we weren't able to maintain the integrity of the HB9:GFP MNs during FACS purification. We also tried to re-plate the EB day6 GFP+ population and the EB day6 GFP- population on separate plates. Unfortunately, with the protocol we use, the motor neuron population alone suffers in culture without the presence of the supporting population, and we were unable to obtain reliable data. To try to answer the reviewer's question, we performed FISH experiments to visualize the expression of IncMN3 in cell culture and saw that after dissociation (DIV1 and DIV7) the IncRNA expression correlates with the GFP signal, confirming its enrichment almost exclusively in GFP+ cells (new Figure 1C). Furthermore, a clear increase in the GFP⁺ population at DIV7 indicates that the observed decrease in IncMN3 expression is due to a dilution effect.

- **The authors claim that IncMN3 expression is enriched in MNs, however the scRNAseq data shown in Fig. S1D IncMN3 is expressed in dorsal spinal cord domains (and ventral to a lesser extent) but actually not in the pMN or MN domains. This statement should be rephrased accordingly.**

According to Fig. EV1D (right panel), we agree with the reviewer that IncMN3 is not expressed in pMN. However, concerning its levels in MNs, the dot plot in the right panel shows that the IncMN3 signal is among the highest in terms of expression levels in this cell population and presents the highest fraction of cells expressing the gene. Thanks for addressing the point, we have highlighted the motoneuronal population for a better comprehension.

- Several sc or snRNAseq studies on human spinal cord have also been recently published (the authors rightly cite one of them, Rayon et al 2021). The authors could also mention if IncMN3 expression was also detected in those studies, and whether it was specially expressed in MNs.

We thank the reviewer for the suggestion, we have added this information to the text along with a new supplementary Figure (EV1E).

- The WB in Fig. S3F does not indicate the age of the mice. Could it be that the prot isn't detected in the SC (or any other mouse tissue) because of a wrong developmental time checked (postnatal or even adult tissues)? (Also, Fig. 3L should be J).

We understand from this comment that the age of mice described in the text was not clear. We have replaced the adjective "adult" with the phrase "three months old" to improve clarity. This should make the data presented more intuitive.

Regarding the expression of the micropeptide, we observed it in the spinal cord of mice at 16.5 days of embryonic development (former Figure 3L, now Figure 3K) and in the spinal cord of 3-months-old mice (former Figure 3I, now Figure 3J), i.e. independently from the developmental stage. Regarding Fig. EV3F, this figure shows the immunoprecipitation performed in lung extracts from three-months-old mice. We performed the same experiments in embryonic lung (E16.5) and obtained the same negative result, suggesting that IncMN3 is not translated in the lung even at a different stage of development. This negative result is reported below.

The panel numbering has been corrected.

- The graphs for Fig 4C-I look odd with some bars showing 3 points, others only 2 and some none. Showing the distribution of all recorded neurons would be better (could be in a violin plot).

Panels D to I of figure 4 do not display all data point because they consist in "box and whisker" plots which, like violin plots, yield the data distribution by showing parameters calculated based on data distribution. In this case the shown parameters are the median value, and 10th and 90th percentile values. Black circles are outlier data falling outside 10th and 90th percentile values. We have modified the legend of fig. 4 to clarify this point.

- In addition, as the authors pointed out, these cultures are normally composed by only ~40% MNs. How did the author know that the recordings were made from MNs and not from any other neural type in the culture?, did they record specifically from HB9:GFP+ cells? If not, the claim that SERTM2 is a major determinant of resting membrane potential in MNs would not be supported, and saying "neurons" would be more appropriate.

Thank you for the comment, we have clarified in the experimental procedures section that the “Electrophysiological experiments were performed on GFP positive motor neurons derived from HB9::GFP-mESCs”

- How do the authors reconcile the results shown in Fig. S4 (no difference in the frequency of spontaneous synaptic currents and spontaneous action potentials across ms ESC-derived neuron genotypes) with the clear differences shown in Fig. 4 on evoked APs? I would have appreciated some mention to this in the discussion.

We have added a paragraph in the Discussion addressing this point. In particular, we now mention that excitability is lowered at depolarized resting membrane potential (in the absence of SERTM2 micropeptide), likely due to a higher percentage of inactivated voltage dependent channels (with a new reference).

- Adding labels to Fig. 5B would help to understand what each molecule represents. Similarly, all WBs should indicate the mw for the proteins shown.

Labels have been added as well as molecular MW.

Dear Dr. Martone,

Thank you for the submission of your revised manuscript. We have now received the enclosed reports from the referees and I am happy to say that both support its publication now. Only a few editorial requests will need to be addressed before we can proceed with the official acceptance of your manuscript:

- The author names on the manuscript title page should be listed as they should be published, i.e. first name and last name.
- The FUNDING info: "National Center for Gene Therapy and Drugbased on RNA Technology" (CN00000041) is missing in our online ms submission system. Please add all funding info when you submit the final ms.
- The author credits need to be removed from the ms text. All credits are entered during online ms submission.
- The 2 figures in the Appendix file can be EV figures, we can now publish more than 5 EV figures. The tables in the Appendix file can be EV tables, so the Appendix file can be deleted. Please update all ms callouts accordingly.
- "Summary" should be renamed to "Abstract"
- The correct order of the manuscript sections is: Abstract, Introduction, Results, Discussion, Methods, Acknowledgements, Disclosure and competing interests statement, References, Figure legends, Expanded View Figure legends

Figure Legends - Comments

- Please note that the box plots need to be defined in terms of minima, maxima, centre, bounds of box and whiskers, and percentile in the legends of figures 4D-I; 6E; EV5 C, G.
- Please note that information related to n is missing in the legends of figures 4C, EV4 G, H; EV5 G.
- Please note that the error bars are not defined in the legends of figures 4C.
- Please note that the white arrow heads are not defined in the legend of figure 1C This needs to be rectified.

I would like to suggest some minor changes to the abstract that needs to be written in present tense:

In this study, we analyze the long non-coding RNA, lncMN3, that is predominantly expressed in motor neurons and shows potential coding capabilities. Utilizing custom antibodies, we demonstrate the production of a lncMN3-derived type I transmembrane micropeptide, SERTM2. Patch clamp experiments performed on both wild-type and SERTM2 knockout motor neurons, differentiated in vitro from mouse embryonic stem cells, show a difference in the resting membrane potential and overall decreased excitability upon SERTM2 depletion. In vivo studies indicate that the absence of the peptide impairs treadmill test performance. At the mechanistic level, we identify a two-pore domain potassium channel, TASK1, known to be a major determinant of the resting membrane potential in motor neurons, as a SERTM2 interactor. Our study characterizes one of the first lncRNA-derived micropeptides involved in neuronal physiology.

EMBO press papers are accompanied online by A) a short (1-2 sentences) summary of the findings and their significance, B) 2-3 bullet points highlighting key results and C) a synopsis image that is exactly 550 pixels wide and 200-600 pixels high (the height is variable). The synopsis image should provide a sketch of the major findings, like a graphical abstract. Please note that text needs to be readable at the final size. Please send us this information along with the final manuscript.

Referee #1:

The authors have addressed all remarks, and I support the manuscript's publication.

Referee #3:

The authors have very satisfactorily addressed all the points I raised. They have done a fantastic job at clarifying my questions and performed the few additional experiments requested (and done some others not even requested). I have no further comments.

All editorial and formatting issues were resolved by the authors.

Dr. Julie Martone
National research council (Italy)
p.le aldo moro 5
Rome, italy 00185
Italy

Dear Dr. Martone,

I am very pleased to accept your manuscript for publication in the next available issue of EMBO reports. Thank you for your contribution to our journal.
